# StarEmbed: Benchmarking Time Series Foundation Models on Astronomical Observations of Variable Stars

## Abstract

Time series foundation models (TSFMs) are increasingly being adopted as highly-capable general-purpose time series representation learners. Although their training corpora are vast, they exclude astronomical time series data. Observations of stars produce peta-scale time series with unique challenges including irregular sampling and heteroskedasticity. We introduce `StarEmbed`, the first public benchmark for rigorous and standardized evaluation of state-of-the-art TSFMs on stellar time series observations ("light curves"). We benchmark on three scientifically-motivated downstream tasks: unsupervised clustering, supervised classification, and out-of-distribution source detection. `StarEmbed` integrates a catalog of expert-vetted labels with multi-variate light curves from the Zwicky Transient Facility, yielding ∼40k hand-labeled light curves spread across seven astrophysical classes. We evaluate the zero-shot representation capabilities of three TSFMs (`Moirai`, `Chronos`, `Chronos-Bolt`) and a domain-specific transformer (`Astromer`) against hand-crafted feature extraction, the long-standing baseline in the astrophysics literature. Our results demonstrate that these TSFMs, especially the `Chronos` models, which are trained on data completely unlike the astronomical observations, can outperform established astrophysics-specific baselines in some tasks and effectively generalize to entirely new data. In particular, TSFMs deliver state-of-the-art performance on our out-of-distribution source detection benchmark. With the first benchmark of TSFMs on astronomical time series data, we test the limits of their generalization and motivate a paradigm shift in time-domain astronomy from using task-specific, fully supervised pipelines toward adopting generic foundation model representations for the analysis of peta-scale datasets from forthcoming observatories.

## 1 Introduction

The adoption of time-series foundation models (TSFMs), with pretraining corpora that span commerce, finance, electricity, and traffic data, is proliferating due to their highly capable, general-purpose representation learning of time-variable signals (Zhou et al., 2021a; Nie et al., 2022; Yang et al., 2024; Woo et al., 2024). TSFMs are not trained on astronomical observations, however, and this omission is consequential because astronomical time series ("light curves") present regimes that are rare in standard benchmarks: multiple variates, irregular time sampling, missing data, and heteroscedasticity (Figure 1). More specifically, there are frequent gaps of variable intervals in the observations (see Figure 1), and the presence of clouds, which change day-to-day and hour-by-hour, yields heteroskedastic uncertainties for the individual observations. At the same time, modern surveys such as the Zwicky Transient Facility (ZTF; Bellm et al., 2019) and the forthcoming Vera C. Rubin Observatory (Ivezić et al., 2019) generate peta-scale volumes of multi-band light curves, creating both a pressing need and a unique opportunity to evaluate TSFM generalization on real scientific data.

Stars that exhibit brightness variations over regular, periodic intervals (periodic variable stars) are astrophysically valuable as they are unique probes of stellar interiors and evolution, galactic structure, and can be used to measure the distance to nearby galaxies (e.g., Feast and Walker, 1987; Clementini et al., 2003; Genovali et al., 2014; Catelan and Smith, 2015; Ripepi et al., 2017). Dozens of types of periodic variable stars exist, and modern astronomical surveys, like ZTF, have produced an avalanche of light curves (∼$10^9$ stars each with ∼$10^3$ observations over 7 yr from ZTF alone). These light

curves are multi-variate because observations are conducted with a filter (or "passband") placed along the focal path of the telescope, limiting the image to only light from a specific wavelength range. Thus, light curves contain both brightness and "color" information (i.e., the relative brightnesses across passbands), enabling inference of physical properties of the source. The abundance of astronomical time series data will dramatically accelerate as the recently-commissioned Vera C. Rubin Observatory (Ivezić et al., 2019) will discover $>10^8$ variable stars while monitoring $>10^{10}$ stars over the course of a decade. Despite this abundance of astronomical light curves, there is no standardized benchmark for assessing time-series embeddings in this domain. The absence of common datasets, class sets, and train-test splits has hindered fair, reproducible comparisons and obscured whether domain-specific pipelines outperform generic representations from foundation models (cf., Pan et al., 2024).

We introduce `StarEmbed`, the first public benchmark for rigorous, standardized evaluation of state-of-the-art (SOTA) TSFMs on astronomical observations. `StarEmbed` integrates expert-vetted labels with multi-band ZTF light curves, yielding ∼40k expert-labeled stars across seven astrophysical classes with fixed train/validation/test splits. To capture the scientific breadth of downstream use, we evaluate three tasks central to time-domain astronomy: unsupervised clustering, supervised classification, and out-of-distribution (OOD) source detection. Our study measures the zero-shot representation quality of three SOTA TSFMs, `Moirai`, `Chronos`, and `Chronos-Bolt`, and a domain-specific transformer (`Astromer`) against the long-standing top-performing baseline of hand-crafted feature extraction that has been widely adopted in the astrophysics literature.

Despite being trained on data completely unlike astronomical light curves, TSFM embeddings, particularly from the `Chronos` family, match or surpass established astrophysics-specific baselines on some tasks and set a new SOTA on our OOD detection benchmark, indicating strong cross-domain transfer and practical utility. These results suggest a possible paradigm shift in astronomy from bespoke, fully supervised pipelines toward generic foundation representations plus lightweight heads to enable petascale time-series analysis for forthcoming observatories.

The major contributions of our work are as follows.

- We introduce the first *standardized* benchmark of time-series foundation models (TSFMs) on astrophysical light curves, revealing the limits and transferability of TSFMs on irregular, heteroscedastic time series.
- We provide evidence for a practical paradigm shift in time-domain astronomy, from bespoke, fully supervised pipelines to off-the-shelf foundation embeddings with lightweight heads, enabling scalable analysis of forthcoming petascale surveys.
- We curate a benchmark dataset of ∼40k expert-labeled ZTF multi-band light curves across seven astrophysical classes, with fixed train/validation/test splits.
- We release embeddings, datasets, code, and detailed documentation to support fair comparison, reproducibility, and future extensions by the community.

The remainder of this paper describes related works and the models we benchmark (Section 2); introduces the ZTF data set (Section 3), provides our benchmark methodology (Section 4), and presents the benchmark results (Section 5) before discussing our concluding thoughts (Section 6).

## 2 RELATED WORKS AND MODELS

Major recent investments in time-domain astronomy have generated incredibly large datasets that naturally lend themselves to machine learning methods. The classification of periodic variable stars has been a problem of significant interest for centuries, as these sources provide direct insight into many facets of stellar astrophysics. As such, both pre- (e.g., Debosscher et al., 2007) and post-deep learning models (e.g., Moreno-Cartagena et al., 2025) have been applied to this problem. We summarize the embedding models and the baseline below. We aim to assess the zero-shot generalization capabilities of the pre-trained TSFMs, so we do not fine-tune them on our data.

### 2.1 SUPERVISED CLASSIFIERS

The first machine learning models to classify variable stars used manually engineered features combined with classical models such as support vector machines (Debosscher et al., 2007) or gradient boosted decision trees (Boone, 2019). Richards et al. (2011) achieved SOTA performance with 52 extracted features (including Fourier coefficients, variability amplitude, skewness, etc.) and

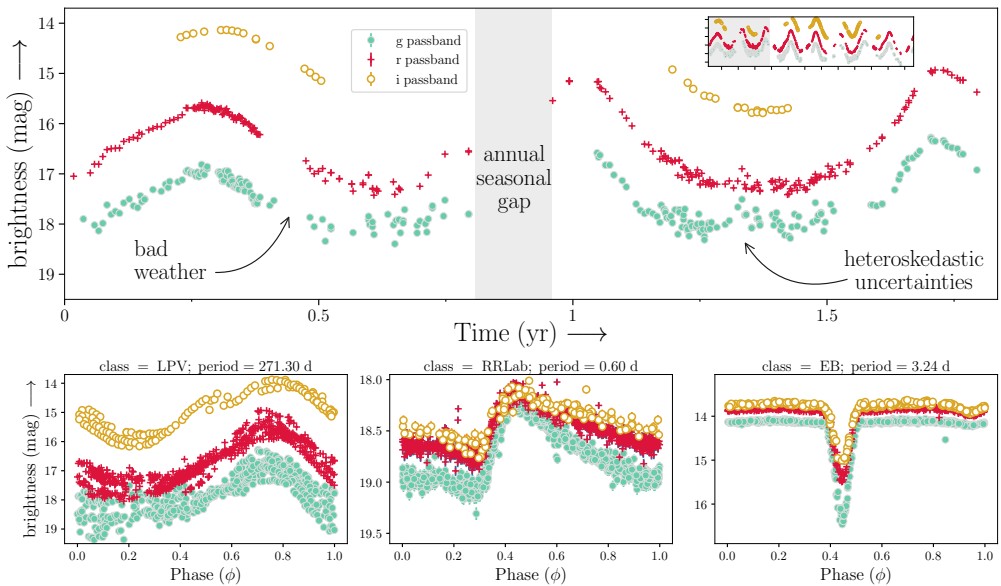

Figure 1: Example ZTF light curves illustrating unique characteristics of astronomical time series, including multiple passbands, large observational gaps, and heteroskedastic uncertainties. *Top panel*: Observed light curve of a periodic variable exhibiting typical characteristics of the observations. The inset shows the full ∼6.5 yr duration of ZTF observations. *Lower panels*: Phase-folded light curves highlighting the differing periodic patterns in three different classes. Note that most stars have few $i$ passband observations so we exclude these data from our analysis (see text for further details).

a random forest (RF) classifier. Later work involved using goodness-of-fit metrics from fitting template physical models to the time series as additional inputs into a tree-based classifier Sesar et al. (2017). Feature extraction varies from study to study, though some have attempted to standardize this step (Nun et al., 2015; Kim and Bailer-Jones, 2016; Malanchev et al., 2021). Recent work has introduced deep-learning methods to eliminate explicit feature engineering using a wide range of architectures including recurrent neural networks (RNNs) (Muthukrishna et al., 2019; Becker et al., 2020; Shah et al., 2025) and transformers (Cabrera-Vives et al., 2024; Moreno-Cartagena et al., 2025). These efforts, however, do not perform meaningfully better than the established hand-crafted feature extraction baseline: accuracies of RNN models are $\pm 1 - 3\%$ of hand-crafted features across multiple variable star datasets (see, e.g., Naul et al., 2018). As a result, we choose to use hand-crafted features to establish our baseline performance.

**Baseline Model:** For this work, we first extract features using the `FATS` (Nun et al., 2015) and `light_curve` (Malanchev et al., 2021) software packages. Example features include: the best-fit Lomb-Scargle (Lomb, 1976; Scargle, 1982) period, the scatter, the skewness, the kurtosis, and other metrics. In total, we define 69 features per passband, yielding a total embedding size of 138 for the two-passband ZTF data (see Appendix C for a full feature list with explanations). We normalize each feature to have zero mean and unit variance. While very effective, hand-crafted features rely heavily on domain knowledge, can be brittle to data quality issues, and are expensive to compute.

## 2.2 ASTROPHYSICS EMBEDDING MODELS

With a high cost to obtain labels for astronomical sources, there has been a growing interest in using semi-supervised approaches to learn general representations of the data to later perform downstream tasks. Recent approaches include variational autoencoders (Villar et al., 2020), sparse autoencoders (Dillmann et al., 2025), and contrastive learning (Zhang et al., 2024), but they are typically limited to a single class (e.g., supernovae). A few foundation models for astronomy attempt to produce useful representations of light curves, such as `FALCO` (Zuo et al., 2025) and `Astromer` (Donoso-Oliva et al., 2023; Donoso-Oliva et al., 2025). Unlike `FALCO`, `Astromer-1` and `Astromer-2` are designed to apply to observations from any observatory (Donoso-Oliva et al., 2023; Donoso-Oliva et al., 2025), and thus, we adopt the `Astromer` models as an astronomy-specific foundation model.

**Astromer** (Donoso-Oliva et al., 2023) is a transformer-based model to generate informative embedded representations of light curves. Astromer-1 was pre-trained using self-supervised learning on 1.5 million single-band light curves from the MACHO survey (Alcock et al., 2000). The model's output is a fixed-length embedding, and, as recommended by the creators, we use the 256-dimensional embedding from the final attention layer produced using the publicly released weights. Astromer-1 was trained to reconstruct masked portions of the input sequence (i.e., masked time series modeling). Astromer-2 (Donoso-Oliva et al., 2025) increases the number of model parameters from 0.66M to 5.4M and adopts an uncertainty-weighted loss function for pretraining. The Astromer models represent the SOTA domain-specific model and serve as a prime candidate for testing whether astronomy-specific pre-training yields discernible benefits relative to general TSFMs.

## 2.3 TIME SERIES FOUNDATION MODELS

TSFMs have been shown to consistently outperform the traditional *one-dataset-per-model* schema in multiple fields, including finance, climate science, and commerce (e.g., Yue et al., 2022; Woo et al., 2024; Ansari et al., 2024). With strong performance that scales with model and data set size, they are a promising tool for driving the future of AI for time series Edwards et al. (2024); Pan et al. (2024). Astronomy, despite having an enormous collection of light curves, has yet to examine the potential of TSFMs, which may prove transformative in our ability to accomplish multiple downstream tasks. Furthermore, large time-domain surveys provide a unique opportunity to evaluate TSFMs with minimal risk of data leakage because astronomical light curves are not included in any of the training corpora. This benchmark therefore provides a new test for how TSFMs transfer to an unseen domain.

**Moirai** (Woo et al., 2024) is designed to be a single foundation model that can forecast virtually any time-series, regardless of sampling frequency, dimensionality, or distribution. It pairs a multi-patch-size projection scheme (i.e., handling minute- to year-scale data), an any-variate attention mechanism that scales to arbitrary numbers of variables, and a flexible mixture-distribution output head for calibrated probabilistic forecasts. Trained on LOTSA (Woo et al., 2024), an open archive of 27 billion observations spanning nine domains, Moirai's Small/Base/Large variants deliver SOTA accuracy in both in-distribution and zero-shot settings, often outperforming models that are fully fine-tuned for a particular dataset.

**Chronos** (Ansari et al., 2024) is another pre-trained time series model showing comparable or even better results than Moirai. It treats forecasting as a language-modeling problem: Chronos scales and quantizes real-valued time-series into a fixed vocabulary, then trains off-the-shelf Transformer language models (T5-style models with 20M to 710M parameters) with an ordinary cross-entropy loss. Augmented by TSMixup and Gaussian-process–generated synthetic data, Chronos is pre-trained on a large collection of public datasets and evaluated on 42 benchmarks. The resulting models deliver strong probabilistic forecasts—significantly ahead of classical and deep-learning baselines on in-domain data and in zero-shot settings, showing that "language of time-series" tokenization alone is enough to build a competitive universal forecaster.

## 2.4 RANDOM EMBEDDINGS AS A SANITY CHECK BASELINE

To establish a performance floor, we generated random vectors as a proxy for light curve embeddings. The 256-length vectors are generated from a $\mathcal{U}[0, 1]$ distribution. The vectors carry no information about the data, meaning this baseline allows confirmation that any alternative models with superior performance capture useful information in the embeddings.

## 3 DATASET

The benchmark dataset includes multi-variate time-series observations of periodic variable stars. The flux is presented in magnitudes (an astronomy specific unit), while the time is recorded as the modified Julian date.

The observations are from ZTF, which repeatedly scans all stars visible from the Northern hemisphere every few days. ZTF observes in three different passbands, $g$, $r$, and $i$[1] (see Figure 1) roughly corresponding to visible green, visible red, and (outside the visible) infrared light, respectively

---

[1]Most ZTF sources have very few or no observations in $i$ band and we therefore exclude it from our analysis.

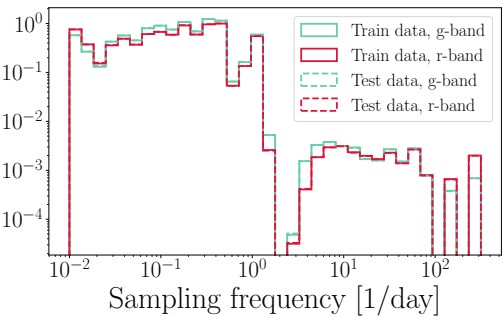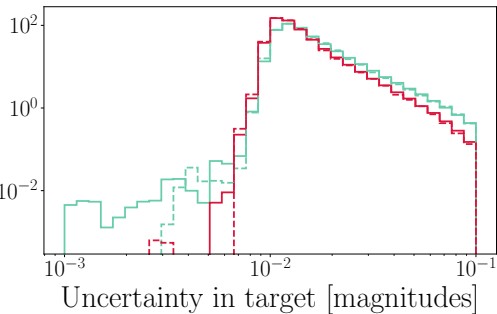

Figure 2: Statistical analysis of our train and test split data. *Left*: histogram of sampling intervals (days between consecutive measurements). *Right*: histogram of observational uncertainties. Sampling frequency varies across >4 orders of magnitude, reflecting the significant challenge of irregular sampling in our light curves. Uncertainties on the target also vary across multiple orders of magnitude. These features are critical properties of light curves which are rarely to never seen in TSFM pretraining data sets.

(Dekany et al., 2020). We use observations from ZTF data release 23 (DR23) which spans a duration of ~6.5 yr and contains billions of light curves. Figure 2 shows a statistical analysis of this data set, highlighting the challenges inherent to these astronomical time series. The left panel shows the sampling frequency or the distribution of time elapsed between two measurements of a given star. This demonstrates the significant irregularity in the temporal sampling of this data. The sparsity at high frequency sampling reflects the limited scientific utility of very high frequency observing, and the gap at frequencies of two per day reflects the fact that observing from the ground can only occur at nighttime and it is always daytime 0.5 days after an observation is taken. The sampling frequency is very consistent across the $g$ and $r$ bands and across the train and test splits. The right panel shows the distribution of measured magnitude uncertainties. Many factors can dramatically affect these uncertainties, and this is reflected in their large spread. We observe that the $g$ and $r$ band uncertainty distributions are slightly different; $r$ band uncertainties are more centrally concentrated and $g$ band uncertainties have greater extremes. We also observe a slight over-density of $g$ band measurements with very small uncertainties in the train split. Noting the logarithmic scale on the y-axis, this is not a concern as these represent only a very small number of observations.

We also note that these features are in stark contrast to those in time series data sets commonly used for TSFM pretraining. LOTSA Woo et al. (2024), e.g., only includes time series without spread in their sampling frequencies, i.e. only regularly sampled time series, and nearly all time series in LOTSA do not include uncertainties on the target or multiple variates. Therefore, this astronomical time series data represents a new, special out-of-domain test for the TSFMs we explore here.

While many labels for periodic variable stars exist within the literature, the vast majority of these labels are derived from low-capacity machine learning models. A careful selection of light curves is therefore warranted to prevent significant label noise within the benchmark. We thus avoid these catalogs in establishing this benchmark.

For training, we instead adopt the Catalina Surveys Periodic Variable Star Catalog (CSPVS; Drake et al., 2014a), a human-labeled catalog of periodic variables discovered by the Catalina Real-Time Transient Survey (Drake et al., 2009). We extract ZTF light curves for each CSPVS star. Stars that lack a ZTF light curve are omitted; we also remove (i) observations flagged as "bad" in ZTF DR23; (ii) light curves with $< 32$ total observations; (iii) light curves that lack both $g$ and $r$ observations; and (iv) light curves from classes with fewer than 350 total examples. The resulting dataset contains ~40,000 ZTF light curves of expert-labeled periodic variable stars across seven classes. See Appendix D for a detailed astrophysical description of each class.

Nature naturally produces an imbalance between the number of periodic variables in different classes, which is further exacerbated by each class having a different detection efficiency (e.g., LPVs have large amplitude variations making them easy to identify). To ensure that each split gets a representative number of examples from each class, we sample each class into the train, validation, and test splits independently in a 7:1:2 ratio. Table 1 shows the counts of examples per class in each split. We release our train-validation-test dataset splits and the generated embeddings

Table 1: Number of periodic variable stars in our dataset across each class and in each split

| Class | EW | EA | RRab | RRc | RRd | RS CVn | LPV |
|---|---|---|---|---|---|---|---|
| Train | 18998 | 2889 | 1386 | 3233 | 298 | 942 | 255 |
| Validation | 2690 | 410 | 194 | 463 | 42 | 134 | 35 |
| Test | 5387 | 818 | 397 | 926 | 83 | 276 | 70 |
| Total after cuts | 27075 | 4117 | 1977 | 4622 | 423 | 1352 | 360 |

on an anonymous public dataset on HuggingFace (`https://huggingface.co/datasets/123anonymous123/StarEmbed`).

## 3.1 ZTF Dataset in Context: Relevance to Upcoming Observatories

Variable star science has an expansive scope that extends beyond ZTF and periodic variables; the `StarEmbed` benchmark is designed to allow for the addition of new datasets and metrics in future expansions. This flexibility is crucial to the long-term health of this benchmark as numerous new time-domain surveys, like LSST, will begin in the coming years. Each new survey has unique observational capabilities and priorities that will affect the resulting embeddings and downstream task performance. As the largest astronomical time-domain experiment to date, ZTF is an apt choice for building a preparatory benchmark dataset that has already been used to explore emerging areas like multi-modality (e.g., Duev and van der Walt, 2021; Carrasco-Davis et al., 2021; Gagliano et al., 2023; Rehemtulla et al., 2024) and transformers (e.g., Allam et al., 2023; Zhang et al., 2024).

All current and future datasets associated with the `StarEmbed` benchmark are public or will be made public. ZTF DR23 data can be accessed through the Caltech Infrared Processing and Analysis Center[2]; the CSPVS catalog is available via VizieR[3] (Drake et al., 2014b); and our selection of ZTF light curves for CSPVS stars will be available on Hugging Face. Our publicly available dataset includes the necessary metadata and a permissive license for reuse. No personal or sensitive information is present in these datasets (they consist of astronomical observations).

## 4 Evaluation Methodology

We assess the quality of embeddings for (1) unsupervised clustering, (2) supervised classification, and (3) out-of-distribution source detection using the embeddings as features. Together, these give a comprehensive view of the intrinsic structure captured by the embeddings, their usefulness for downstream tasks, and provide a unique generalization benchmark for TSFMs. Below we detail the experimental settings including training procedures and metrics used to evaluate the embeddings. We release our code to reproduce our benchmark experiments (`https://tinyurl.com/jwew993p`). To maintain consistency throughout the benchmark, we use identical embedding sizes across different models whenever possible (i.e., when there exists such a pre-trained version of the model). `Astromer-1`, `Astromer-2`, `Chronos-Bolt-Tiny`, `Chronos-Tiny` and the Random Embeddings all have embedding size of 256. We adopt the smallest available pre-trained `Moirai` model, `Moirai-small`, which uses an embedding size of 384.

## 4.1 Unsupervised Clustering

In this setting, we treat the embeddings of each model as points in a feature space and apply clustering algorithms to see if natural groupings correspond to known variable star classes. Specifically, we use K-means clustering with $k = 7$ corresponding to the number of true classes in the dataset. Before executing the clustering algorithms, we normalize all embeddings to the standard normal because the clustering methods compute Euclidean distances which are sensitive to the scale of entries.

We produce uncertainties on performance metrics by repeating the K-means algorithm with 10 different initializations, choosing the clustering with the lowest within-cluster variance. We also apply Ward's hierarchical clustering, which optimizes the same within-cluster variance objective via agglomerative merges. It provides a robustness check since it is deterministic and initialization-free.

---

[2]`https://irsa.ipac.caltech.edu/Missions/ztf.html`
[3]`https://cdsarc.cds.unistra.fr/viz-bin/cat/J/ApJS/213/9`

We then evaluate clustering quality using the following literature-standard metrics for clustering (Huang et al., 2020; Monnier et al., 2020; Sun et al., 2024; Li et al., 2024):

**Normalized Mutual Information (NMI)**: NMI measures the mutual information between the cluster assignments and the true class labels, normalized to the range [0,1]. An NMI of 1 indicates perfect correlation between clusters and classes, while an NMI near 0 indicates no better than random assignment. NMI is invariant to label permutations, which is suitable since cluster labels are arbitrary.

**Adjusted Rand Index (ARI)**: ARI evaluates pairwise clustering agreements. It considers how often pairs of light curves are in the same cluster vs. the same true class. An ARI of 1 indicates perfect clustering, ARI≈0 indicates random clustering, and ARI<0 indicates clustering worse than random.

**Macro-averaged F1 score (F1)**: Macro-F1 is the harmonic mean of the completeness (true positive rate) and purity (the false positive rate subtracted from unity) computed per class and then averaged evenly across all classes. Because it treats each class equally, the Macro-averaged F1 is sensitive to performance on minority classes. This is especially important for our CSPVS dataset because some classes (e.g., RRd) have many fewer labels than others (e.g., EW; see Table 1). Similarly to previous work (e.g., Monnier et al., 2020), we assign clusters to class predictions using the Hungarian matching algorithm (Kuhn, 1955). This treats clustering as unsupervised classification and makes results directly comparable to supervised settings.

With NMI and ARI, we can asses which embeddings have more separable class structure without any supervised training. Large NMI/ARI scores suggest that the embedding has useful information for differentiating the variable star classes. We also visualize the embedding spaces with dimensionality reduction via a uniform manifold approximation and projection (UMAP) to provide an intuitive, qualitative view of clustering performance in Appendix B.

## 4.2 SUPERVISED CLASSIFICATION

To directly measure how useful the embeddings are for classifying variable stars, we train four simple classifiers on the fixed embeddings to predict the variable star class labels. This simulates a scenario where one uses a pretrained model to produce embeddings used as feature inputs for a classification task but does not fine-tune the embedding model (hence "zero-shot" in terms of the embedding model). We evaluate the embeddings with four (simple) classifiers: a non-parametric model ($k$-nearest neighbor, $k$-NN), a linear probe, a decision tree classifier (random forest, RF), and a non-linear model (multilayer perceptron, MLP). Both $k$-NN and linear probes are standard in the embedding evaluation literature (Caron et al., 2021; Zhou et al., 2021b; Neelakantan et al., 2022), as they are simple methods that directly reflect separability in the embedding space. RF is included, in part, because it is widely used in the astronomical literature for periodic variable star classification (Naul et al., 2018; Sánchez-Sáez et al., 2021; Pimentel et al., 2022) and it achieves SOTA performance across many datasets (Naul et al., 2018). Finally, an MLP is used as a modern higher capacity deep-learning option. Detailed information on the classifiers, the hyperparameter optimization, and the final hyperparameters for each embedding model can be found in Appendix A. We report standard multi-class classification metrics to comprehensively assess the performance of each embedding model on the downstream supervised classification task. *Accuracy*: the fraction of stars that are correctly classified. *Macro-averaged F1 Score (F1)*: The Macro-averaged F1 (see Sec. 4.1) is apt because it is sensitive to performance on minority classes. *Precision/Recall*: The overall precision and recall are included to provide a fully comprehensive evaluation of the classification performance.

## 4.3 OUT-OF-DISTRIBUTION SOURCE DETECTION

Identifying variable stars physically unlike those in labeled training sets is of great astrophysical-interest. To test the effectiveness of the embeddings for detecting such OOD sources, we compute OOD scores for light curves with a modified isolation forest algorithm (Gupta et al., 2025). We first create embeddings for the ZTF light curves of CSPVS stars with too few examples to be included in the training set ($\beta$-Lyrae, Blazhko, Anomalous Cepheids, Cepheid-II, HADS, LADS, ELL, Hump, PCEB, and EAup; see Sec. 3). We define these as OOD sources. The embeddings of the OOD sources are mixed with the test set and run through a "multi-class isolation forest" (Gupta et al., 2025) where a separate isolation forest is fit to the embeddings of each of the seven inlier classes in the training set. The minimum of the seven isolation forest scores is the OOD score we use for OOD source detection. Isolation forest is a popular method for finding astrophysical outliers (Malanchev et al., 2021), and Gupta et al. (2025) show that following this multi-class prescription yields better macro-averaged

Table 2: Results of unsupervised clustering with K-means and Ward. The best results are highlighted in **bold**, and the second-best results are underlined. The `Chronos` models perform very well on this unseen data, placing first or second in all metrics and universally better than `Morai-small` and the `Astromer` models. However, hand-crafted features perform the best overall.

| Methods | K-means | | | Ward (Hierarchical) | | |
|---|---|---|---|---|---|---|
| | NMI | ARI | F1 | NMI | ARI | F1 |
| `Astromer-1` | 0.0041(0.0001) | 0.0017(0.0011) | 0.1660(0.0014) | 0.0041 | 0.0001 | 0.1652 |
| `Astromer-2` | 0.0082(0.0010) | 0.0192(0.0078) | 0.1590(0.0042) | 0.0091 | 0.0310 | 0.1600 |
| `Moirai-small` | 0.1749(0.0017) | 0.0981(0.0028) | 0.2831(0.0034) | 0.1476 | 0.0828 | 0.2612 |
| `Chronos-tiny` | 0.2374(0.0082) | **0.1596(0.0029)** | 0.3110(0.0362) | 0.1890 | 0.1217 | **0.3671** |
| `Chronos-Bolt-tiny` | 0.2120(0.0033) | 0.1306(0.0125) | 0.3128(0.0027) | 0.2273 | **0.1553** | 0.3662 |
| Random Embeddings | 0.0003(0.0001) | 0.0000(0.0000) | 0.0977(0.0007) | 0.0003 | 0.0004 | 0.1122 |
| Hand-crafted Features | **0.2700(0.0058)** | 0.1197(0.0092) | **0.3960(0.0271)** | **0.2508** | 0.1319 | 0.3323 |

performance than a single isolation forest in many settings, including for periodic variables. Here, the performance is benchmarked with the fraction of sources in the top $N^{\text{th}}$ percentile of OOD scores which are genuine OOD sources: the OOD purity.

## 5 BENCHMARK RESULTS

We highlight the benchmark results below for each of the three downstream tasks: unsupervised clustering, supervised classification, and OOD detection.

### 5.1 UNSUPERVISED CLUSTERING

In Table 2 we show that TSFMs generally perform well: (i) the first or second best ranked model in each metric comes from the `Chronos` models; (ii) both `Chronos` models always outperform `Moirai-small`; and (iii) `Chronos-tiny` outperforms `Chronos-Bolt-tiny` on four of our six metrics. Still, we find that the hand-crafted features yield the best overall performance. We also observe that the pre-trained domain-specific `Astromer` models generally yield poor performance, notably always worse than the TSFMs which are not trained on light curves. We further analyze the poor performance of `Astromer-1` in Appendix E and find it's embeddings of our ZTF light curves have collapsed to similar directions. Appendix E further discusses how poor performance is expected for both `Astromer` models based on results from previous studies. Finally, hand-crafted features achieve the highest global separability (top NMI under both K-means and Ward), reflecting coarse class alignment. In contrast, `Chronos-tiny` leads on pairwise consistency (best ARI for K-means, and ARI is more sensitive to pairwise correctness), suggesting that its embeddings form small, pure neighborhoods rather than single, class-wide clusters.

### 5.2 SUPERVISED CLASSIFICATION

Table 3 and the left panel of Figure 3 show that (i) the `Chronos` models once again perform very well compared to `Moirai-small` and the `Astromer` models; (ii) `Astromer-2` performs better than `Moirai-small` in some metrics; and (iii) unlike in the clustering results, `Chronos-tiny` outperforms `Chronos-Bolt-tiny` and achieves the second best performance in nearly all metrics. As in the clustering results, the hand-crafted features are clearly superior, yielding a F1 score of $0.804 \pm 0.003$ with the RF classifier. We also find that the RF classifier generally performs better than others, although this is somewhat model-dependent. We conduct a feature importance analysis on the hand-crafted features in Appendix F and find that the period of the variability in either passband are the most important features.

The center and right panels of Figure 3 show the confusion matrix of one of the best performing TSFM-classifier pairings (`Chronos-tiny` with the MLP) and best overall performing model-classifier pairing (hand-crafted features with an RF). These confusion matrices show that (i) both `Chronos-tiny` and the hand-crafted features often confuse RRd sources as RRc; (ii) `Chronos-tiny` yields better performance on most classes (EA, RRd, RS CVn, and LPV) although loses overall due to the larger margins in the classes where the hand-crafted features perform better (EW, RRab, RRc). In general, these results show that, despite having never seen astronomical time series, `Chronos-tiny` clearly

Table 3: Results of supervised classification across classifiers and embedding models. The best results are highlighted in **bold**, and the second-best results are underlined. The $k$-NN and logistic classifiers are deterministic so only the 1-run performance is reported; the RF and MLP are run with 10 seeds and we report the mean and standard deviation of these runs. The hand-crafted features are state-of-the-art with `Chronos-tiny` a clear second-best.

| Classifier | Metric | Astromer-1 | Astromer-2 | Moirai-small | Chronos-tiny | Chronos-Bolt | Random | HF |
|---|---|---|---|---|---|---|---|---|
| $k$-NN | Accuracy | 0.644 | 0.823 | 0.809 | 0.857 | 0.807 | 0.648 | **0.881** |
| | Precision | 0.130 | 0.660 | 0.662 | 0.799 | 0.647 | 0.120 | **0.818** |
| | Recall | 0.141 | 0.489 | 0.509 | 0.623 | 0.542 | 0.140 | **0.661** |
| | F1 | 0.122 | 0.537 | 0.554 | 0.672 | 0.570 | 0.120 | **0.712** |
| logistic | Accuracy | 0.073 | 0.648 | 0.705 | 0.750 | 0.709 | 0.094 | **0.838** |
| | Precision | 0.147 | 0.486 | 0.544 | 0.575 | 0.549 | 0.144 | **0.663** |
| | Recall | 0.165 | 0.668 | 0.680 | 0.730 | 0.676 | 0.128 | **0.854** |
| | F1 | 0.072 | 0.521 | 0.579 | 0.617 | 0.580 | 0.076 | **0.714** |
| RF | Accuracy | 0.676 (0.000) | 0.846 (0.000) | 0.823 (0.001) | 0.862 (0.000) | 0.826 (0.001) | 0.676 (0.000) | **0.920 (0.001)** |
| | Precision | 0.111 (0.043) | 0.799 (0.006) | 0.716 (0.007) | 0.750 (0.056) | 0.707 (0.002) | 0.097 (0.000) | **0.866 (0.003)** |
| | Recall | 0.143 (0.000) | 0.526 (0.002) | 0.514 (0.002) | 0.597 (0.002) | 0.548 (0.001) | 0.143 (0.000) | **0.773 (0.004)** |
| | F1 | 0.115 (0.000) | 0.580 (0.002) | 0.557 (0.002) | 0.638 (0.002) | 0.582 (0.001) | 0.115 (0.000) | **0.804 (0.003)** |
| MLP | Accuracy | 0.446 (0.147) | 0.627 (0.037) | 0.717 (0.031) | 0.783 (0.022) | 0.721 (0.022) | 0.308 (0.203) | **0.833 (0.022)** |
| | Precision | 0.154 (0.006) | 0.453 (0.019) | 0.546 (0.022) | 0.589 (0.024) | 0.553 (0.026) | 0.137 (0.006) | **0.672 (0.025)** |
| | Recall | 0.165 (0.003) | 0.627 (0.020) | 0.722 (0.006) | 0.758 (0.006) | 0.696 (0.013) | 0.145 (0.002) | **0.851 (0.009)** |
| | F1 | 0.138 (0.020) | 0.470 (0.023) | 0.594 (0.019) | 0.643 (0.023) | 0.589 (0.015) | 0.094 (0.044) | **0.723 (0.027)** |

extracts useful information from the data for supervised classification. The complete set of confusion matrices across all embedding–classifier combinations is presented in Appendix I.

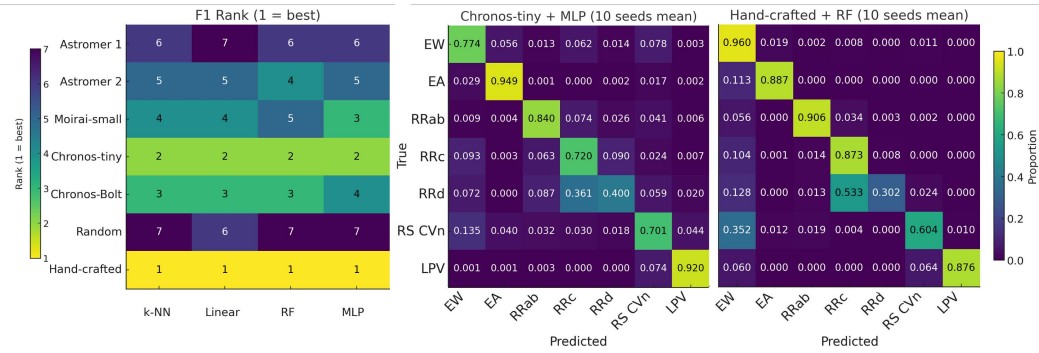

Figure 3: *Left:* F1 Ranking across all baselines with different classifier heads. The `Chronos-tiny` model consistently outperforms other TSFMs and the domain-specific `Astromer` models, but the hand-crafted features provide the best overall performance. *Right:* Confusion matrix of `Chronos-tiny` + MLP, one of the best performing TSFM-classifier combinations, and the confusion matrix of hand-crafted features with the RF classification, the SOTA baseline in astrophysics. `Chronos-tiny` yields better performance on most classes (EA, RRd, RS CVn, and LPV), indicating that the TSFM is effectively extracting appropriate information for classification.

## 5.3 OUT-OF-DISTRIBUTION SOURCE DETECTION

Table 4 shows that: (i) `Chronos-Bolt-tiny` is exceptional at isolating OOD sources from the inliers; (ii) by comparison, hand-crafted features deliver a much lower OOD purity; and (iii) every other model we tested provides only a marginal gain, if any, over evaluating the whole dataset. Of all the sources evaluated for this test, $\sim 11\%$ are OOD samples. This implies that by applying the MCIF approach to the `Chronos-Bolt-tiny` embeddings, we would be able to recover nearly half of the OOD sources, by evaluating just $10\%$ of the data, a $\sim 5\times$ improvement in search efficiency over the random embeddings. As these OOD events often correspond to astrophysically rare or anomalous sources, they are of great interest for the astrophysical community. To interpret the performance of `Chronos-Bolt-tiny` model, we hypothesize that `Chronos-Bolt-tiny`'s patch-based, multi-step objective is less sensitive to step-level variation than `Chronos`'s autoregressive next-token training, hence encouraging a tighter inlier manifold and larger off-manifold distances for rare morphologies. This yields weaker clustering of inliers but stronger OOD isolation. A more careful analysis is left to future work. While these results show promise, it's worth noting that $50\%$ purity implies that we would still need an expert in the loop to vet candidates flagged by such a system.

Table 4: Results for out of distribution source detection. The best results are highlighted in bold, and the second-best results are underlined. The `Chronos-Bolt-tiny` performs very well on this task, ranking first across all metrics with hand-crafted features being a distant second.

| | Purity | | |
| --- | --- | --- | --- |
| | **Top 1 percentile** | **Top 5 percentile** | **Top 10 percentile** |
| Astromer-1 | 0.017(0.016) | 0.091(0.019) | 0.120(0.001) |
| Astromer-2 | 0.135(0.027) | 0.126(0.006) | 0.120(0.002) |
| Moirai-small | 0.169(0.024) | 0.143(0.007) | 0.150(0.004) |
| Chronos-tiny | 0.139(0.054) | 0.116(0.028) | 0.149(0.021) |
| Chronos-Bolt-tiny | **0.569(0.060)** | **0.532(0.055)** | **0.519(0.038)** |
| Random Embeddings | 0.116 | 0.116 | 0.116 |
| Hand-crafted features | 0.213(0.013) | 0.271(0.014) | 0.259(0.003) |

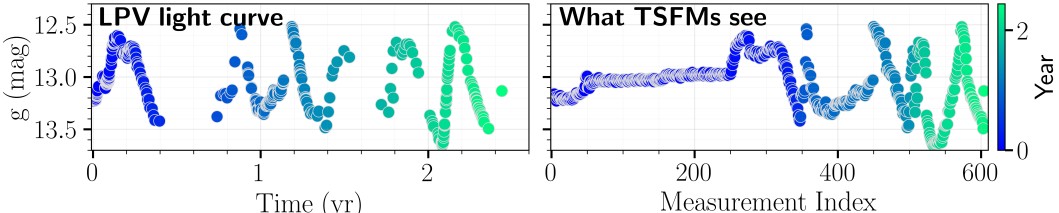

Figure 4: A long period variable (LPV) star's light curve indexed by the measurement timestamps (*left*: the real, physical index) and indexed by the order of the measurements (*right*: a non-physical index). In both panels, the coloring corresponds to the physical time index, showing that discarding relative timestamp information causes the TSFMs to receive a warped, view of the light curve, limiting their interpretation of the data.

## 6 DISCUSSION AND CONCLUSIONS

We introduced `StarEmbed`, a public benchmark for evaluating TSFMs on real multi-band stellar light curves, whose irregular sampling and heteroskedasticity differ substantially from typical TSFM pretraining corpora. By harmonizing expert-vetted CSPVS labels with ~40,000 ZTF light curves, we provide a rigorously curated seven-class dataset and evaluate three key tasks: unsupervised clustering, variable-star classification, and OOD detection. Using this benchmark, we compare (i) domain-specific embeddings (`Astromer`); (ii) SOTA general-purpose TSFMs (`Moirai` and `Chronos`); and (iii) hand-crafted features. Three findings emerge: (i) for clustering, `Chronos` matches hand-crafted features; (ii) for supervised classification, hand-crafted features perform best, with `Chronos` consistently second by a small margin; and (iii) for OOD detection, `Chronos-Bolt-tiny` significantly outperforms hand-crafted features. Across all tasks, TSFMs generally outperform `Astromer` in zero-shot.

Taken together, our results demonstrate the boundaries of TSFM generalization. Figure 4 illustrates a key limitation of TSFMs: not properly treating irregularly sampled data results in a significantly warped view of the time series. This warped view destroys key information like the period of the variability which we found to be the most important feature driving the performance of our strong baseline (Appendix F). TSFMs which (i) introduce architectural mechanisms to treat irregularly sampled time series and (ii) include such data in their pretraining corpora will alleviate this problem and likely excel in our benchmark. In astronomy, such a model would drive a paradigm shift from bespoke, fully-supervised pipelines toward off-the-shelf foundational representations plus lightweight task heads for variable star analysis. These frameworks would push analysis frontiers in the, now ongoing, era of astronomy driven by observatories creating petabyte-scale datasets (Ivezić et al., 2019). In response, astronomy has challenging, information-rich time series in game-changing quantities to offer the AI for time series field. Making use of these time series would also help open new applications of TSFMs beyond astronomy as other scientific domains also produce irregularly sampled time series. By releasing all data, code, and model wrappers, `StarEmbed` serves to forge the bridge between astronomy and the AI for time series fields and to provide the community with a benchmark for foundation model advances on challenging time series data.

## ETHICS STATEMENT

We have reviewed the Code of Ethics and affirm that all authors have read it, adhere to it, and that this submission complies with its requirements.

## LLM USAGE DISCLOSURE

We use large language models (LLMs) to aid with polishing writing including improving clarity and grammar. We also use LLMs to search for works related to those described here. All the technical results are original contributions by the authors.

## REPRODUCIBILITY STATEMENT

We release all code required to reproduce our results, including dataset construction (Section 3), embedding generation, training with hyperparameter optimization and three-run scripts, evaluation routines, plotting utilities, and supporting bash scripts. All the code are provided in the supplementary materials and the following anonymous link: `https://tinyurl.com/jwew993p`. We host a anonymous public dataset on HuggingFace for our curated benchmark dataset and all the produced embeddings (`https://huggingface.co/datasets/123anonymous123/StarEmbed`).

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

A​PPENDIX

## A DETAILS OF EXPERIMENTS FOR CLASSIFICATION

We report the hyperparameter tuning process and summary for all classifiers in this section. To fairly compare the different embeddings, we conduct a hyperparameter search on each model when training the downstream MLP and random forest classifier. We use MLP with three hidden layers of sizes 1024, 512 and 256 and an output layer for class predictions. we search over batch size $B \in \{128, 256, 512, 1024\}$, learning rate $lr \in \{0.01, 0.001, 0.0001\}$ and dropout rate $\in \{0.0, 0.1\}$. Every hyperparameter triple runs once on NVIDIA H100 4 GPUs. The training process is at most 50 epochs, and stops early if the validation loss fails to improve for 3 epochs. In practice this training takes less than 30 epochs for all models before the early stopping is triggered. For random forest, we search over maximum depth of the tree $\in \{None, 10, 20, 30\}$, the minimum number of samples to split an internal node $\in \{2, 5, 10\}$, and number of estimator $\in \{100, 200, 500\}$. We summarize the hyperparameters of the MLP and random forest classifiers in Table 5 and 6. For linear classifier, since the current training set is relatively small, we use the `LogisticRegression` (L-BFGS) from Scikit-learn library (Buitinck et al., 2013), with `max_iter = 5000`, `class_weight = "balanced"`, and all other with default settings. L-BFGS is a deterministic full-batch method that converges to the global minimizer without learning rate tuning. For $k$-NN, we use `KNeighborsClassifier` from Scikit-learn with default settings. Since $k$-NN and the default solver of logistic regression are both deterministic, we only report the 1 run result.

Table 5: Best hyper-parameters for each model with the MLP classifier

| Method | Hyperparameters | Training epochs |
|---|---|---|
| Astromer-1 | batch_size=32, learning_rate=0.0001, dropout=0.0 | 17 |
| Astromer-2 | batch_size=32, learning_rate=0.0001, dropout=0.0 | 23 |
| Moirai-small | batch_size=64, learning_rate=0.001, dropout=0.0 | 11 |
| Chronos-tiny | batch_size=32, learning_rate=0.0001, dropout=0.0 | 26 |
| Chronos-Bolt-tiny | batch_size=128, learning_rate=0.0001, dropout=0.1 | 17 |
| Random Embeddings | batch_size=128, learning_rate=0.0001, dropout=0.0 | 5 |
| Hand-crafted features | batch_size=32, learning_rate=0.0001, dropout=0.1 | 30 |

Table 6: Best hyperparameters for each model with the random forest classifier

| Method | Hyper-parameters | Training time (s) |
|---|---|---|
| Astromer-1 | max_depth=10, min_samples_split=2, n_estimators=200 | 84 |
| Astromer-2 | max_depth=30, min_samples_split=10, n_estimators=500 | 456 |
| Moirai-small | max_depth=None, min_samples_split=2, n_estimators=500 | 738 |
| Chronos-tiny | max_depth=None, min_samples_split=5, n_estimators=100 | 126 |
| Chronos-Bolt-tiny | max_depth=30, min_samples_split=2, n_estimators=500 | 450 |
| Random Embeddings | max_depth=None, min_samples_split=2, n_estimators=100 | 198 |
| Hand-crafted features | max_depth=None, min_samples_split=10, n_estimators=100 | 36.4 |

## B VISUALIZATIONS OF EMBEDDINGS

We include the UMAP visualizations for the embeddings from each embedding models to provide more intuitions regarding the embedding space. As shown by Figure 5, all time series pretrained models' embeddings are showing clear distinction and distribution of different clusters corresponding to different ground truth classes. In comparison, as a baseline, the random embeddings show no clear clusters at all. `Astromer-1`'s embeddings and hand crafted features do not show clear clusters either. `Astromer-2`'s embeddings show clearer cluster distribution but for some classes, the clusters are not distinctive with others either. These UMAP visualizations further demonstrate the promising potentials of using time series pretrained models as light curve embedding models.

## C FULL LIST OF HAND-CRAFTED FEATURES

We select hand-crafted features from the libraries of established software packages: `FATS` (Nun et al., 2015) and `light_curve` (Malanchev et al., 2021). Each feature described here is computed for each passband individually and the embeddings are formed by concatenating the feature lists of the

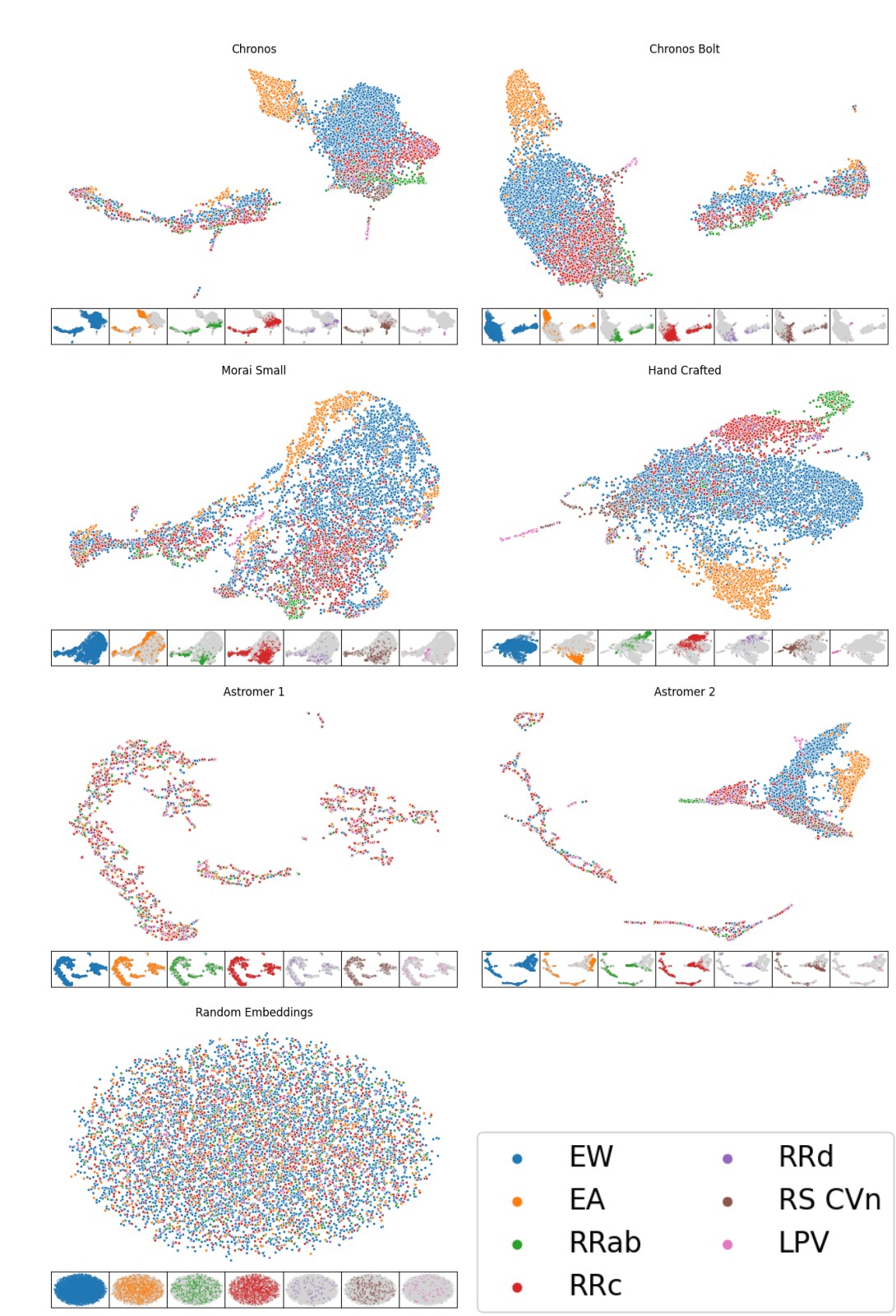

Figure 5: UMAP projections for each embedding model included in our analysis using the test set. Inset plots at the bottom of each figure show clustering of different classes.

Table 7: FATS features and plain-language descriptions.

| Feature | Intuitive one-sentence description |
|---|---|
| PeriodLS | Best-fit period of the light curve using the Lomb–Scargle method. |
| Period_fit | The false alarm probability of the largest Lomb–Scargle periodogram value. |
| Psi_CS | The range of a cumulative sum metric computed on the phase-folded light curve. |
| Psi_eta | The variability index $\eta^e$ computed on the phase-folded light curve. |
| Autocor_length | The cross-correlation of the light curve with itself. |
| PairSlopeTrend | The fraction of increasing first differences subtracted from the fraction of decreasing first differences, computed on the 30 most recent magnitude measurements. |
| Freq{N}_harmonics_amplitude_{M} | Amplitude of the $M$th harmonic of the $N$th dominant frequency. |
| Freq{N}_harmonics_rel_phase_{M} | Relative phase of the $M$th harmonic of the $N$th dominant frequency. |
| CAR_sigma | Short-term variability amplitude in a continuous auto-regressive (CAR) model. |
| CAR_tau | Characteristic timescale of correlations in the CAR model. |
| CAR_mean | Long-term mean magnitude level in the CAR model. |

See here for a detailed description: `http://isadoranun.github.io/tsfeat/FeaturesDocumentation.html`

$g$ and $r$ embeddings. Tables 7 and 8 show the full list of features and descriptions from FATS and `light_curve`, respectively.

## D  ASTROPHYSICAL DESCRIPTION OF CLASSES

Our dataset contains seven total classes of periodic variable stars: EW, EA, RRab, RRc, RRd, RS CVn, and LPV. Here, we provide a high-level astrophysical description of each of these classes, including each class' observational characteristics and utility. In some cases, multiple classes are closely related so we describe them together. We also include descriptions of the classes which, due to their rarity, are considered out-of-distribution in this work: $\beta$-Lyrae, Blazhko, Anomalous Cepheids, Cepheid-II, HADS, LADS, ELL, Hump, PCEB, and EAup.

### D.1  ECLIPSING BINARIES (EW, EA)

Eclipsing binary stars are pairs of stars orbiting each other and aligned with the observer in such a way that either star periodically blocks the light from the other. When neither star is eclipsed, the system is at maximum brightness, but when one star is eclipsed by the other, the total flux received from the system is suppressed, giving the binary star system periodic light curve behavior. This type of variability is described as extrinsic because it is not due to astrophysical properties of the stars themselves. Sub-categorization of eclipsing binaries is based on the configuration of the stars in the pair.

EW-type eclipsing binaries (also called W Ursae Majoris-type after the original EW system) are contact binaries. In this case, the two stars, typically dwarf stars, share a common outer envelope which entirely encapsulates them. This common envelope allows for the exchange of mass and energy between the pair, equilibrating their temperature. Their light curves exhibit constant and smooth variations where the dips from either star being eclipsed are of similar or identical depth. EW-type variables typically have short periods ($0.2 \lesssim P \,[\mathrm{d}] \lesssim 0.5$), with a notably unsolved period cut-off at $\sim 0.2$ days (Rucinski, 2007; Drake et al., 2014a). These systems are astrophysically valuable, in part, because they are expected to emit gravitational waves due to their tight orbits, and they also have the possibility of merging and triggering transient events (Tylenda et al., 2011).

Table 8: `light_curve` features and plain-language descriptions

| Feature | Intuitive one-sentence description |
|---------|-----------------------------------|
| Amplitude | Half the peak-to-peak range—how far the light curve swings between brightest and faintest points. |
| AndersonDarlingNormal | Scores how strongly the magnitude distribution departs from an ideal bell curve. |
| BeyondNStd | Proportion of data points that sit more than $N$ standard deviations away from the mean, flagging outliers. |
| Cusum | Total vertical span of the running cumulative sum, revealing slow drifts or trends. |
| Eta | Von Neumann ratio: compares successive-point differences to overall scatter to catch rapid variability. |
| EtaE | Eta re-weighted by time gaps so uneven sampling doesn't skew the variability estimate. |
| InterPercentileRange($p$) | Distance between the $p$ and $(1-p)$ quantiles—a robust width such as the IQR (when $p$=0.25). |
| Kurtosis | Indicates whether the distribution is more peaked or heavy-tailed than a normal curve. |
| LinearFit | Slope, error, and fit quality for a straight line that accounts for measurement uncertainties. |
| LinearTrend | Slope and error of a simple least-squares line that ignores the error bars. |
| MagnitudePercentageRatio | Ratio of inner to outer percentile widths, contrasting core spread with overall spread. |
| MaximumSlope | Steepest single-step change in magnitude per unit time between consecutive points. |
| Mean | Ordinary average magnitude. |
| Median | Mid-point magnitude that splits the data into equal halves. |
| MedianAbsoluteDeviation | Typical absolute distance from the median—a robust scatter measure. |
| MedianBufferRangePercentage | Fraction of points that fall inside a narrow buffer zone around the median. |
| OtsuSplit | Statistics describing the two groups produced by Otsu's automatic thresholding of magnitudes. |
| PercentAmplitude | Largest absolute deviation of any point from the median magnitude. |
| ReducedChi2 | Reduced $\chi^2$ showing how well the data match their (weighted) mean given the quoted errors. |
| Skew | Tells whether the distribution leans toward brighter or fainter extremes (positive or negative tail). |
| StandardDeviation | Classical root-mean-square scatter of the magnitudes. |
| StetsonK | Error-weighted "peakedness" measure that is robust to outliers in light-curve shape. |
| WeightedMean | Average magnitude that gives greater weight to points with smaller measurement errors. |

See here for a detailed description: `https://github.com/light-curve/light-curve-python`

EA-type eclipsing binaries (also called Algol-type binaries after the original EA system) are detached binaries. In this case, the two stars are not in contact and thus can have different temperatures and more varied orbits, manifesting as different light curve properties. The ellipticity of the orbit and the brightnesses of the stars affect the spacing and depths of the brightness dips. Multiple additional factors can affect their light curves, e.g., the presence of an accretion disk. EA systems tend to have longer periods than EW systems due to their wider orbits: $0.3 \lesssim P$ [d] $\lesssim 100$. EA systems are key for studying binary stellar evolution and populations, especially the exchange of mass between stars in a binary.

### D.2 Active Binaries (RS CVn)

RS Canum Venaticorum (RS CVn) stars are also stars in binary systems and are characterized by one of the stars exhibiting large magnetic spots on its surface. This manifests as observable variability as the RS CVn stars also have rapid rotational velocities. The resulting light curve affect also depends on the difference between the star's rotational period and the systems orbital period, and most RS CVn systems are tidally locked, meaning the two periods are closely matched. Some RS CVn are also eclipsing binaries, so their light curves can also show variability due to eclipses. Their periods tend to be $3 \lesssim P$ [d] $\lesssim 14$. RS CVn systems serve as extreme testbeds for studying stellar magnetic phenomena and evolution.

### D.3 RR Lyrae (RRab, RRc, RRd)

RR Lyrae stars are low-mass stars exhibiting pulsations, cyclically expanding and contracting radially due to internal changes in opacity. Because RR Lyrae occur with only a small range of intrinsic brightnesses, their distances can be easily measured from their observed brightness. Among other utilities, this allows RR Lyrae to be used for measuring distances within the Milky Way and to nearby Galaxies. Their periods are also related to their chemical composition, so they can provide crucial information about Galactic structure and formation.

RR Lyrae can occur in different pulsation modes which define the various RR Lyrae subclasses. RRab stars pulsate in the fundamental mode; RRc in the first-overtone; and RRd are double-mode pulsators. RRab stars have light curves with a rapid brightening episode followed by a gradual fading, producing a sawtooth-like pattern. RRab typically have periods $0.4 \lesssim P$ [d] $\lesssim 1.0$. RRc stars exhibit sinusoid-like variability with typical periods of $0.2 \lesssim P$ [d] $\lesssim 0.5$. They also tend to exhibit a constant, slow drift in their periods. RRd stars pulsate in both the fundamental mode and the first-overtone and thus show a combination of two periodic signals in their light curves, typically with the first-overtone dominating.

### D.4 Long Period Variables (LPVs)

LPVs are giant stars exhibiting pulsations with periods of $3 \lesssim P$ [d] $\lesssim 1000$. They include multiple subtypes each with different period-luminosity relations but we, as in Drake et al. (2014a), consider these a single class. They pulsate with a similar mechanism to the RR Lyrae but have dramatically larger radii. Their outer layers are not very tightly bound to the rest of the star, which can lead to the star expunging mass and polluting the surrounding environment with gas. Thus, studying LPVs provides insights into the cycles of gas into and out of stars. Their period-luminosity relations also allow LPVs to act as distance measures.

### D.5 Beta-Lyrae

Beta-Lyrae ($\beta$-Lyrae) stars are close binary star systems in which the outer gaseous layers of both stars have expanded to the point that the pair is enveloped in a shared gaseous envelope. At this stage, gaseous material can be transferred from one star to another, altering either star's evolution. Their periods are typically a few days, and their light curves display continuous variations in brightness rather than flat maxima or minima like EA-type binaries, for example. Their dynamics provide direct information on gaseous mass transfer between binary stars, stellar structure under extreme tidal distortion, and the role of binarity in late stellar evolution.

### D.6 Blazhko

Blazhko variables are a sub-class of RR Lyrae variables, which exhibit the rare Blazhko effect in which their light curve amplitudes and phases are modulated over long time periods (tens to hundreds of days). In part owing to its rarity relative to normal RR Lyrae stars, there is not yet a consensus to the physical mechanism driving the Blazhko effect. The effect may be explained with magnetic fields or resonances within the star, so these stars are useful for studying exotic phenomena which can occur in stars.

### D.7 Type II Cepheids (Cepheid-II) and Anomalous Cepheids (ACEP)

Cepheid-II stars have old, low-mass stars with periods typically of tens of days and can produce a variety of light curve morphologies. They deviate from classical Cepheids because they are much fainter, but they do follow their own period-luminosity relation, enabling them to also be used as distance measures.

ACEPs have periods and luminosities inbetween those of RR Lyrae and classical Cepheids (roughly 0.3–2 days) with amplitudes of about 0.3–1.0 magnitudes, and their light curves typically resemble those of RRab stars. Their physical nature is not very well understood, but they have be proposed to be a product of gaseous mass transfer in a binary star system. ACEPs provide special astrophysical insights into stellar evolution pathways involving binary interaction, as well as into the environments where they typically occur.

### D.8 High-amplitude Delta-Scutis (HADS) and Low-amplitude Delta-Scutis (LADS)

Delta-Scuti ($\delta$-Scuti) stars are pulsating variables stars with short periods and are typically divided into the HADS and LADS subclasses based on the morphology of their light curves. HADS have simple, regular, sawtooth-like light curves with periods <0.3 days and amplitudes greater than ~0.3 magnitudes. In contrast, LADS exhibit complex, multi-periodic light curves with amplitudes below ~0.1 magnitudes. HADS provide clean tests of stellar pulsation theory and scaling relations, while LADS are testbeds for asteroseismology.

### D.9 Ellipsoidal Binaries (ELL)

ELLs are close binary star systems in which the stars are tidally distorted into ellipsoidal shapes, producing photometric variability without eclipses. ELL light curves have smooth, nearly sinusoidal variations with two unequal minima per cycle, arising from the elongated parts rotating in and out of view. They are astrophysically important because they reveal details of binary star evolution, stellar shapes, and the presence of companion objects such as white dwarfs, neutron stars, or black holes.

### D.10 Hump variables

The Hump class is used as a catch-all for the small amount of periodic variables which Drake et al. (2014a) were unable to classify into any other known classes but do show clear periodic variability. Some of these objects exhibit vaguely sawtooth-like variability, like what is seen in RRab stars, but others have smoother variability.

### D.11 Post-Common-Envelope Binaries (PCEB)

PCEBs are close binary stars that have recently emerged from a common-envelope evolutionary phase, in which one star expanded and engulfed its companion star inside its expanded outer layers ("envelope"). Their light curves show a wide range of morphologies, including eclipses and ellipsoidal modulations, depending on the physical properties of the system like the inclination of the orbit relative to our line-of-sight and the nature of either of the stars in the binary system. Their periods tend to be very short as the common-envelope phase drives angular momentum out of the orbit. The smaller star in these systems are often a white dwarf, so these systems are an opportunity to study potential progenitors of Type Ia supernovae.

### D.12 EA with unknown period (EAup)

EAup are EA-type stars where Drake et al. (2014a) were unable to determine their periods for any reason.

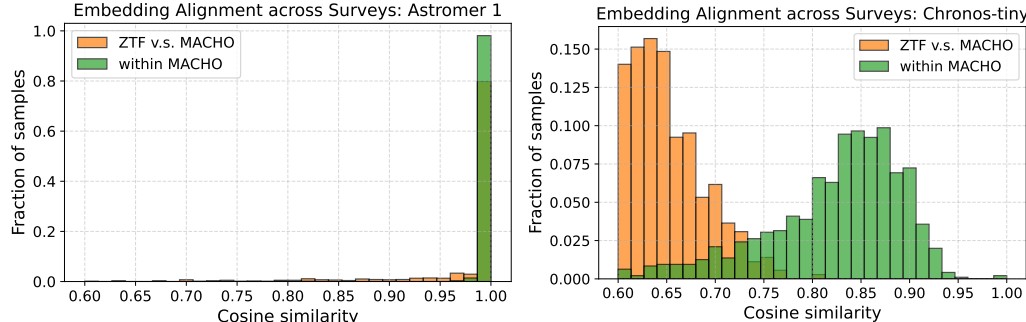

Figure 6: **Comparison of Embedding Elignment for `Astromer-1` (left) and `Chronos-tiny` (right) models.** The plots show the distribution of cosine similarities between light curve embeddings, both within the MACHO survey (green) and across surveys (ZTF vs. MACHO, orange). `Astromer-1` exhibits embedding collapse, with cosine similarities approaching 1.0 no matter within- or cross-survey. This indicates that the model encodes little discriminative structure. In contrast, `Chronos-tiny` produces more meaningful embeddings. The wider distribution of cosine similarities preserves structural information, and the clear separation between within-survey and cross-survey pairs demonstrates its ability to capture the domain shift between datasets.

## E  ASTROMER PERFORMANCE AND ASTROMER-1 EMBEDDING QUALITY

We provide detailed analysis and provide experiments on the issue of the poor performance of `Astromer-1`. First, `Astromer-1` needs further finetuning on the dataset of the downstream task to achieve good performance on the variable star classification task, according to Donoso-Oliva et al. (2023). This is evidenced in their Fig. 11 (a), which shows a clear increase of F1 score from  0.25 to 0.6 when finetuned on 20 to 500 variable stars per class of the MACHO dataset. A similar trend is observed for other datasets, including OGLE-III and ATLAS. `Astromer-2` (Donoso-Oliva et al., 2025) also shows improvement with finetuning, though its performance starts higher (around 0.65) even with just 20 samples per class. Please refer to (Donoso-Oliva et al., 2025, Figs. 11 and 12) for the details. These results indicate that `Astromer`'s low zero-shot performance in our benchmark is expected, as we intentionally evaluate the pretrained checkpoints without any task-specific tuning.

Second, we provides empirical analysis to show that `Astromer-1` embedding collapse into similar direction. Specifically, we randomly sample 1000 pairs of ZTF–MACHO data in r-band, and compute the cosine similarity of embedding from two model, `Astromer-1` and `Chronos-tiny`. For comparison, we do the same for within-MACHO pairs. The result is in Figure 6. For `Astromer-1`, it is clear that its embedding collapse to one direction. Even the 10th percentile of cosine similarity of embedding within MACHO is 0.995, this means every star is almost parallel to each other. Even if the survey data shift, the 10th percentile is still 0.948. From above to we conclude the embedding of frozen `Astromer-1` encodes very little discriminative structure. This explain why downstream classifier has a hard time to distinguish different class (low F1 score). For `Chronos-tiny` (right figure), the cosine similarity within MACHO (green) shows a wide range of angles, indicating the embeddings preserve class information. Furthermore, the embedding has a clear domain shift. The cosine similarity of ZTF-MACHO pairs (orange) is lower than the one of within-MACHO pair. Unlike frozen `Astromer-1`, `Chronos-tiny` doesn't collapse everything into a single direction. It still has room to spread out unseen patterns instead of forcing them into the old manifold.

## F  FEATURE IMPORTANCE ANALYSIS FOR HAND-CRAFTED FEATURES

In this appendix we investigate the feature importance of the individual hand-crafted features to investigate the source of their excellent performance. We perform this analysis on the random forest trained for supervised classification with the hand-crafted features because it offers built-in functionality for quantitatively measuring the importance of each feature. The feature importance is computed using the mean decrease in impurity. In this framework, each time a feature is used to split a node, the associated reduction in the Gini impurity metric is recorded and weighted by the number

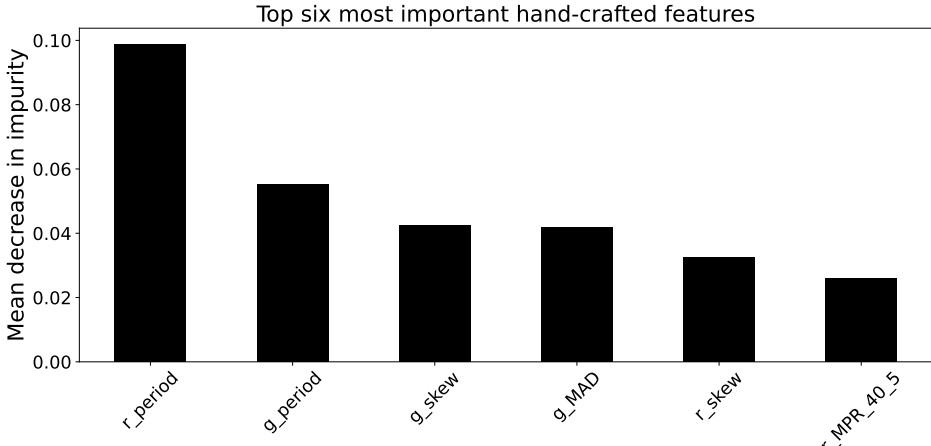

Figure 7: Top six most important hand-crafted features for random forest classification. The period of variability computed for either passband rank the highest. Other important metrics also include the skewness of the distribution of magnitudes, the median absolute deviation, and the magnitude percentage ratio for the 40th and 5th percentiles.

of samples reaching that node. These weighted impurity decreases are then summed over all nodes and all trees in the ensemble, and finally normalized to yield a global, unit-scaled measure of how strongly each feature contributes to improving class separation within the forest.

Figure 7 shows the results of our feature importance analysis. We find that the period inferred from the $g$ and $r$ bands of the time series are the most important features driving the performance of the hand-crafted features. This aligns with expectations as numerous other studies performing variable star classification with tree-based classifiers come to a similar conclusion Richards et al. (2011); Dubath et al. (2011); Kim and Bailer-Jones (2016). We also find that the skewness, median absolute deviation, and the magnitude percentage ratio also rank very highly. These features encode information about the spread of the magnitude measurements, indirectly representing the morphology of the periodicity of the star. All features are described in Appendix C.

## G    PRELIMINARY RESULTS OF FINE-TUNING TSFMs ON ASTROPHYSICS DATA

Our goal in this work is to answer a very specific first-order question: how useful are existing, large TSFMs without any model adaptation for real, irregular astrophysical light curves? This is fundamentally a question of generalization capabilities, motivated by strong claims made by the latest-generation TSFMs (Chronos, Chronos-bolt, Moirai). To this end, we explicitly treat these TSFMs as frozen embedding models and do not fine-tune them on our data. This "zero-shot" setting on new, out-of-domain data is precisely where generalization abilities can be tested.

Methodologically, we do go beyond pure zero-shot prediction by training lightweight supervised heads (k-NN, linear probe, RF, MLP) on top of frozen embeddings (see Sec. 4.2). This isolates the question we care about: do the representations produced by TSFMs, without architecture- or domain-specific fine-tuning, perform competitively with conceptually very simple baselines? Our results show that they do: TSFMs approach or surpass these baselines for clustering and OOD detection and come close for supervised classification, despite never being trained on astronomical data. This alone is a strong and surprising conclusion.

To better understand the potential of adaptation, we also conduct a preliminary, and deliberately simple, fine-tuning experiment on Moirai-small. We fine-tune the model on about 1M ZTF light curves for about 24 hours using the same masked token prediction task as in the pre-training. As shown by Figure 8, the training loss decrease from 1.25 to 1.15 within the first 20k steps but then enter a regime of strong fluctuations around a saturated plateau over the next 180k steps, without clear signs of further improvement. This is the evidence that straightforward, off-the-shelf fine-tuning

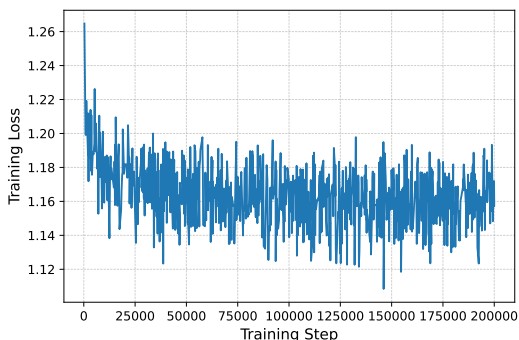

Figure 8: Training loss of fine-tuning Moirai-small on 1 million of ZTF variable star light curves. The train loss saturates quickly after a small improvement and fluctuates around a plateau. Straightforward, off-the-shelf fine-tuning of TSFMs to light curve data requires non-trivial modifications to the model architecture and/or the standard fine-tuning practices and does not immediately yield clear gains over the zero-shot setting.

is not trivial and does not immediately yield clear gains over the zero-shot setting. This result in aligned with our interpretations on the shortcomings of the TSFMs: there exists a significant gap between the irregular, sparse astrophysical light curves and the regular time series on which these TSFMs were originally pre-trained. Not treating the irregular sampling of the data destroys much of the important information it holds, see Fig. 4. Fine-tuning these TSFMs on astronomical time series is a non-trivial, model- and domain-design problem in its own right.

Exploring one/few-shot model adaptation–e.g., careful parameter-efficient fine-tuning (PEFT) of selected layers under small labeled budgets, with appropriate regularization and class balancing– would therefore require substantial additional design choices: which layers to adapt, which PEFT strategy, what label budgets, how to avoid overfitting on common classes, etc. This constitutes a large and interesting future research track for follow-up work of our benchmark.

## H PERIOD REGRESSION RESULTS

As we discuss in Section 6 and Appendix F, our hypothesis on the fact that TSFMs underperform hand-crafted feature is due to the failure of capturing the true period of the light curve. Thus, we have added period regression as an additional task in our benchmark.The result is in Table 9.

| Model | RMSE(d) | $R^2$ |
|---|---|---|
| MOIRAI | $26.355 \pm 0.022$ | $0.295 \pm 0.001$ |
| Chronos-bolt | $23.414 \pm 0.062$ | $0.443 \pm 0.002$ |
| Chronos | $23.178 \pm 0.076$ | $0.455 \pm 0.003$ |
| Astromer-1 | $33.272 \pm 0.015$ | $0.005 \pm 0.0009$ |
| Random-emb | $33.452 \pm 0.016$ | $-0.005 \pm 0.0009$ |

Table 9: Period regression on the model embedding. While differences do exist in the regression performance of these embeddings, it is important to note that they all perform signficantly worse than Lomb Scargle periodogram which is a reliable, albeit computational expensive, method used for period finding in Astronomy. Note that we exclude hand-crafted features since they already contain the period.

# I    ADDITIONAL EXPERIMENTS RESULTS ON CONFUSION MATRIX

Figure 9: Confusion matrix of `Chronos-tiny` on four classifiers.

To provide detailed information about classification performance across different variable star classes, we report the confusion matrices for all embeddings and the hand-crafted features in this section. For the random forest and MLP classifiers, the per-class accuracies are obtained by averaging over the results of 10 runs.

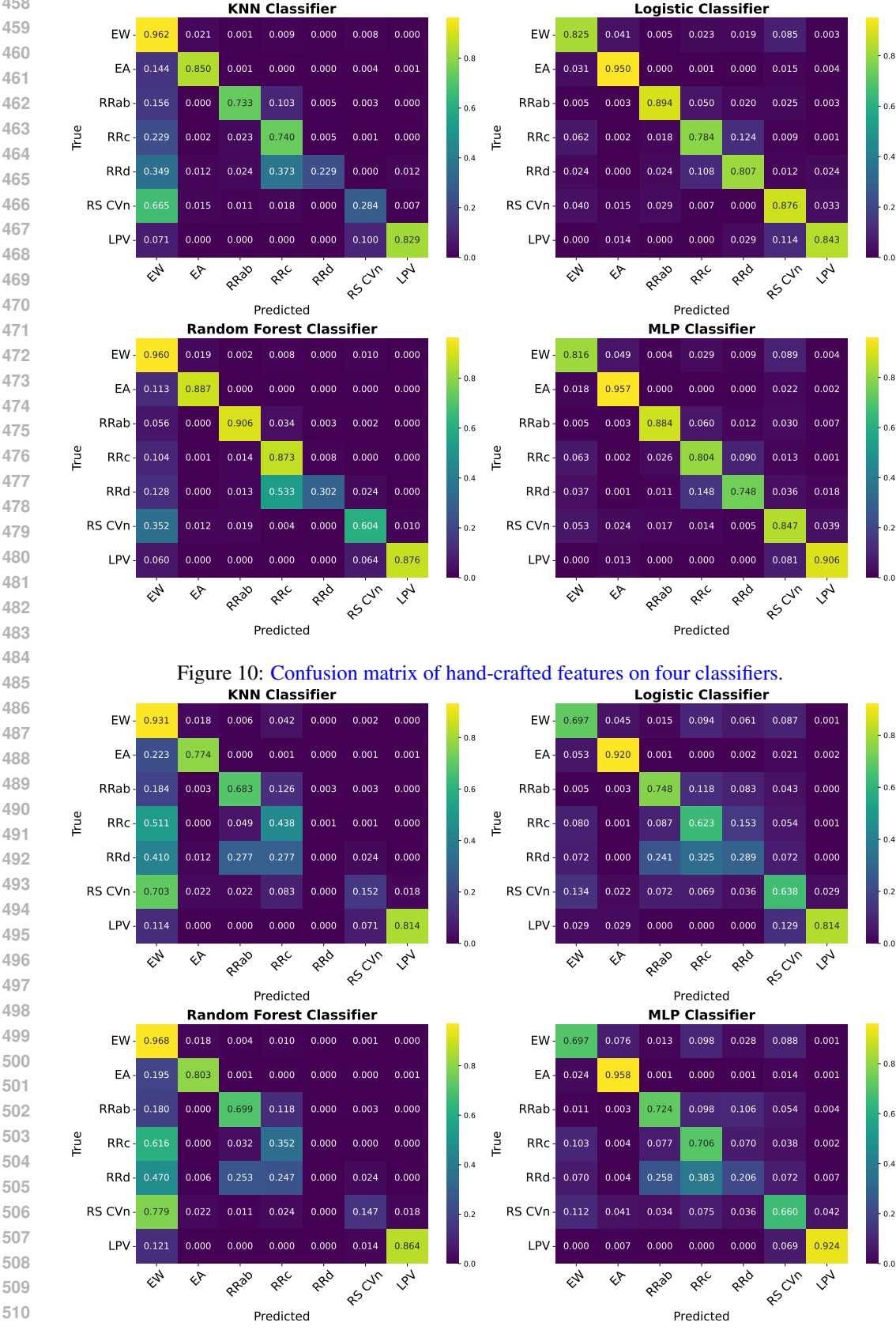

Figure 10: Confusion matrix of hand-crafted features on four classifiers.

Figure 11: Confusion matrix of `Chronos-Bolt` on four classifiers.

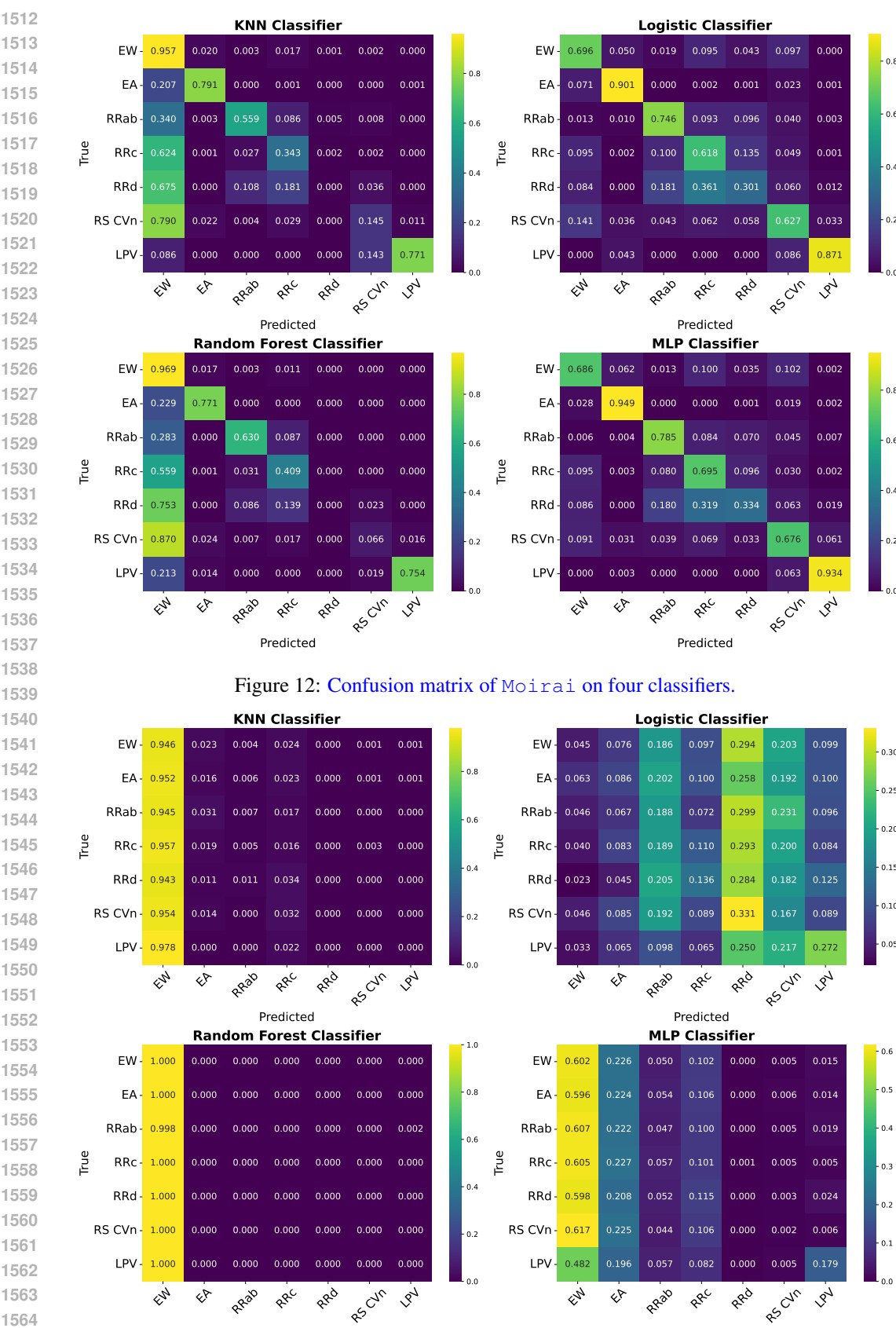

Figure 12: Confusion matrix of `Moirai` on four classifiers.

Figure 13: Confusion matrix of `Astromer-1` on four classifiers.

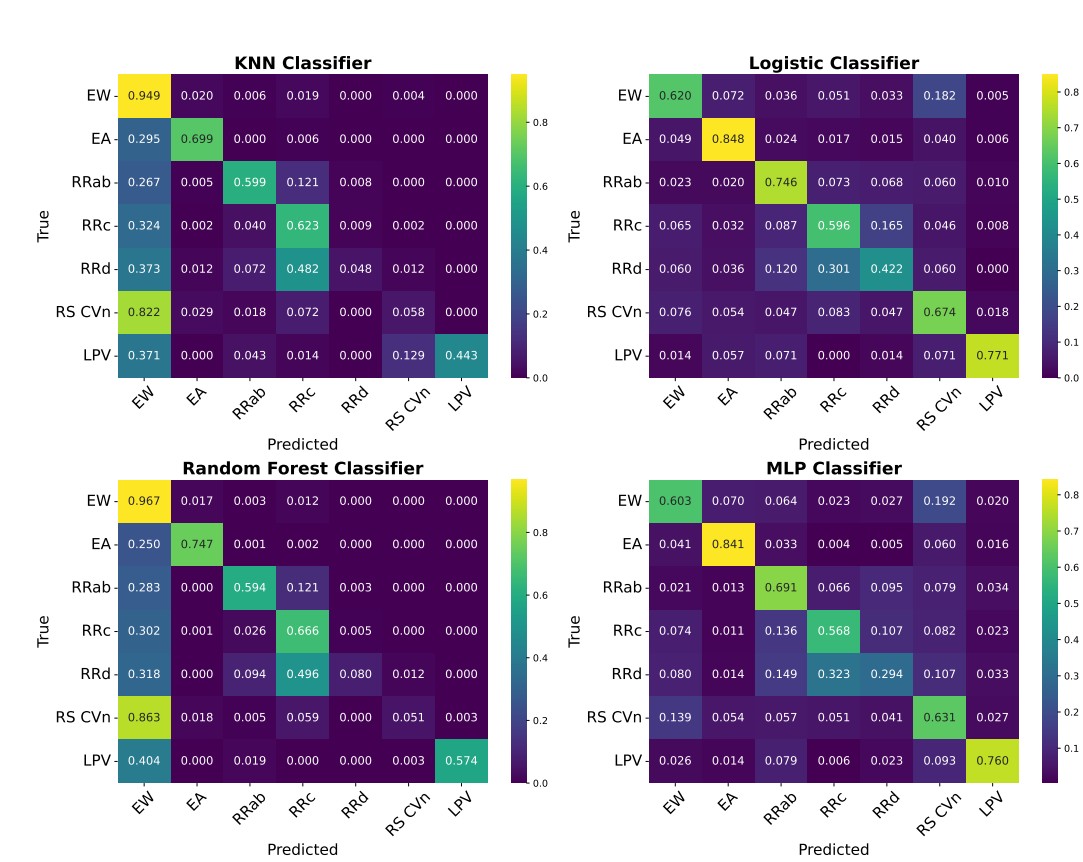

Figure 14: Confusion matrix of `Astromer-2` on four classifiers.

