# OpenReview forum: "StarEmbed: Benchmarking Time Series Foundation Models on Astronomical Observations of Variable Stars"
_ICLR.cc/2026/Conference — ICLR 2026 Conference Withdrawn Submission_

### Official Review · Reviewer_xzSZ · 2025-10-29

**Soundness:** 2
**Presentation:** 3
**Contribution:** 2
**Rating:** 4
**Confidence:** 3

**Summary:**

This paper evaluates the performance of time series foundation models in astronomy science. The authors introduce a new benchmark dataset (StarEmbed) comprising over 40,000 hand-labeled light curves spanning seven astrophysical classes. On this new dataset, the authors conducted unsupervised clustering, supervised classification, and out-of-distribution source detection tests. The results demonstrate that although these TSFMs were not trained on astronomical observation data, they can outperform existing astrophysics-specific models in certain tasks. Particularly noteworthy is the exceptional performance of TSFMs in out-of-distribution detection, where they significantly surpass domain-specific models. The authors emphasize that these experimental findings are driving a paradigm shift in astronomy data analysis.

**Strengths:**

- The paper is well written and easy to follow.
- It presents a qualified benchmark study: introducing a new dataset which is very important for time series research, performing diverse experiments on it, and revealing several noteworthy experimental results.

**Weaknesses:**

- The paper's technical contribution is limited.

**Questions:**

- StarEmbed is a time series classification benchmark. Why did the authors choose to test Chronos and Moirai, which are designed for forecasting, rather than specialized time series classification foundation models in the paper?

---

> ### Author Response · Authors · 2025-11-22
> **Part 1 of 1: Technical Contribution and Other TSFMs**
>
> Thank you for taking the time to review our submission and for providing thoughtful feedback. We appreciate your efforts and the opportunity to clarify several points raised in your review. The following is our response, and the revision includes corresponding updates (marked in BLUE).
>
> >The paper's technical contribution is limited.
>
> **Response**: Thank you for raising this point. We agree that the original submission did not sufficiently highlight the technical contributions beyond benchmarking. In the revised manuscript, we have **expanded both the analyses and the discussion** to clarify the insights our benchmark provides. We summarize these additions in our responses to `reviewer bdhm` (part 2, part 3) and `reviewer qUQx` (part 1), where we outline the new diagnostics, analyses, and findings introduced in the revision, as well as the contribution to the community.
>
> Concretely, we now (i) add new experimental analyses (feature-importance on hand-crafted features, dataset-level statistics, and a period-regression task), (ii) provide a clearer hypothesis and evidence for where current TSFMs succeed and where they fail on irregular, heteroskedastic scientific time series, and (iii) include an fine-tuning experiment to illustrate that naive adaptation of Moirai is non-trivial.
>
> We hope these clarifications make the contribution and its technical value more apparent.
>
> ---
>
> > StarEmbed is a time series classification benchmark. Why did the authors choose to test Chronos and Moirai, which are designed for forecasting, rather than specialized time series classification foundation models in the paper?
>
> **Response**: We appreciate the reviewer’s question. Our central goal is to evaluate off-the-shelf time-series foundation models in a strict zero-shot setting on irregular astrophysical light curves, not to propose new architectures. To the best of our knowledge, there is currently no widely adopted, pre-trained “time series classification foundation model” that (i) is trained on large, diverse corpora, (ii) is publicly available, and (iii) can be used in the same plug-and-play zero-shot manner as Chronos and Moirai (i.e., frozen encoder producing general-purpose embeddings). If the reviewer is aware of such models that meet these criteria, we would be grateful for specific pointers and would be happy to reference them as related work.
>
> Furthermore, we highlight that it is standard practice to use a foundation model as a frozen encoder to produce an embedding for different downstream tasks, even when the models are trained to predict the masked token/pixel. Chronos and Moirai, although originally framed as forecasting models, are pre-trained exactly in the same way: via masked-token prediction or autoregressive sequence modeling over massive, heterogeneous time-series corpora. This mirrors the way BERT-style and GPT-style language models are trained and reused as general-purpose encoders, where their intermediate embeddings are successfully applied to downstream classification tasks without changing the backbone. In our work, we follow this well-established practice (see the citations in `section 4.2`, `line 354-360` in the revision): we treat Chronos and Moirai as frozen encoders, extract embeddings, and attach simple classifier heads (k-NN, linear, RF, MLP) to evaluate their utility for variable-star classification, clustering, and OOD detection.
>
> Thus, while Chronos and Moirai were marketed primarily as forecasters, our results demonstrate that their learned representations are also competitive for classification on challenging, irregular astrophysical time series—often approaching or surpassing long-standing hand-crafted baselines. This is precisely one of the key messages of our paper: that large, pre-trained TSFMs, even when originally developed for forecasting, can serve as powerful, general-purpose representation learners for scientific time-series analysis.
>
> ---
>
> Thanks for your time and thoughtful review. We hope our clarification addresses your question. If there are any remaining concerns, we would be happy to discuss further!

---

### Official Review · Reviewer_qUQx · 2025-10-30

**Soundness:** 2
**Presentation:** 2
**Contribution:** 2
**Rating:** 4
**Confidence:** 3

**Summary:**

This paper introduces StarEmbed, a benchmark designed to evaluate state-of-the-art time series models on stellar time series data. The authors assess the models on three tasks: unsupervised clustering, supervised classification, and out-of-distribution source detection. The results show that Chronos models perform particularly well, especially in the out-of-distribution source detection task.

**Strengths:**

1. A benchmark is always a good contribution to the community to encourage and push the frontier of models and comparing on a far basis the algorithm

2. The authors make an effort on the experimental side to perform host of experiments.

**Weaknesses:**

1. The paper's contribution feels limited. It mainly introduces a benchmark and tests existing models, without offering a novel architecture, fine-tuning, or a new foundational model based on Chronos. A more substantial contribution, such as a new model or an innovative approach (e.g., boosting/bagging), would make the work more impactful and valuable to the community. Since Chronos is already well-known, testing it on the benchmark alone doesn't constitute a major contribution. While introducing the benchmark is still valuable, it’s not enough on its own to make the paper stand out.

2. If the focus is on the benchmark, more effort should have gone into its statistical analysis. This includes examining the distribution across samples, channels, and time, studying distribution shifts, and analyzing the train/test split to understand the benchmark’s difficulty and challenges. Frequency statistics and two-dimensional visualizations could help illustrate the benchmark's significance. Additionally, including more baseline models for comparison such as Toto, Moment, or PatchTST would strengthen the paper’s contribution and provide a more comprehensive evaluation.

**Questions:**

1. Did the authors perform any statistical analysis of the benchmark, such as examining the distribution across samples, time, and channels? It would also be useful to explore frequency diversity, wavelet decomposition, and PCA/t-SNE representations to better understand the data.

2. How is the train/test split performed, and what unique challenges does it present? Is there meaningful signal in the benchmark, or is it just noise? The authors should discuss whether the benchmark contains realistic, non-trivial information and offer insights into its significance.

3. It would be beneficial to test the benchmark on other foundational models from the literature, as well as models commonly used in time series forecasting, to provide a more comprehensive evaluation.

---

> ### Author Response · Authors · 2025-11-22
> **Part 1 of 4: Contribution of Benchmark**
>
> Thank you for taking the time to review our submission and for providing thoughtful feedback. We appreciate your efforts and the opportunity to clarify several points raised in your review. Below we provide our responses, and the revision includes corresponding updates (marked in BLUE).
>
> > The paper mainly introduces a benchmark and tests existing models, without offering a novel architecture, fine-tuning, or a new foundational model based on Chronos. A more substantial contribution, such as a new model or an innovative approach (e.g., boosting/bagging), would make the work more impactful and valuable to the community.
>
> **Response**: We refer the reviewer to the detailed response in **Parts 2 and 3 to `reviewer bdhm`** where we provide additional analysis and insights into TSFMs to clarify the utility of our benchmark.
>
> Below is a short summary of those parts:
> * first benchmark to evaluate general-purpose TSFMs on astronomical time series, with characteristic challenges not seen in data sets currently used by TSFM practitioners
> * TSFMs generalize much better than domain-specific transformer-based models despite having never seen astronomical time series
> * Current SOTA TSFMs critically lack the ability to properly treat irregularly sampled time series, and alleviating this holds promise for significant advancements
>
> **On the need for “novel architecture / new foundational model/fine-tuning.**
> We fully agree that designing improved models is an important research direction. However, doing so requires exactly the kind of benchmark and analysis we provide: without a standardized dataset, clear baselines, and a diagnosis of current failure modes, it is difficult to know which design choices matter and what to improve. Furthermore, **training a new foundational model with novel architecture also requires substantial compute and data-engineering effort** (try various data mixtures/filtering strategies;) that is well beyond the scope of this benchmark paper. At the same time, we agree that fine-tuning results can further strengthen the contribution. As a result, **we now include a Moirai fine-tuning experiment in the revised manuscript.** See the detailed fine-tuning result and discussion on the response to Part 2 to `reviewer bckc`, and `Appendix G` in the revision.
>
> Lastly, this benchmark also holds significant value for the astrophysics community as well as potential for advancing the connection between AI for time series and other scientific communities. The tasks and metrics we design are reflective of wide-reaching and active research in the astrophysics literature. These evaluations provide the astrophysics community with clearer-than-ever insight for discerning which model is best-suited for their novel analysis. We add text in `Section 6` discussing how the challenges reflected in our astronomical time series are shared with those in other scientific domains, so StarEmbed represents an early effort to bridge these disparate communities and advance them together.

---

> ### Author Response · Authors · 2025-11-22
> **Part 2 of 4: Statistical Analysis of Data Set**
>
> > More effort should have gone into [the benchmark's] statistical analysis. This includes examining the distribution across samples, channels, and time, studying distribution shifts, and analyzing the train/test split to understand the benchmark’s difficulty and challenges. Frequency statistics and two-dimensional visualizations could help illustrate the benchmark's significance.
>
> **Response**: Indeed, a statistical analysis of the dataset we present is critical to the robustness of our benchmark. **We have added significant text and a figure reflecting this to Section 3.** There, we show the distribution of sampling frequencies and target measurement uncertainties for the g and r bands as well as the train and test splits. This figure shows a number of critical properties of our data set: (i) irregular sampling is made clear with non-monotonic variations in sampling frequency across four orders of magnitude; (ii) uncertainties also vary across multiple orders of magnitude and show slightly different distributions for the g and r variates. These key properties are nearly entirely absent from the time series in LOTSA, which TSFMs like Moirai are pre-trained on.

---

> ### Author Response · Authors · 2025-11-22
> **Part 3 of 4: Train/Val/Test Split and Realism of Benchmark**
>
> > How is the train/test split performed
>
> **Response**: Performing the train/val/test splits in a robust manner is important for a benchmark-focused study like ours. We separate the data into the three splits by randomly selecting 70%/10%/20% of the data from each class to go to train/validation/test. It is important to note that we did this **splitting independently by class because this ensures we preserve the class imbalance inherent to the dataset**. This is described in the text in Section 3.
>
> > What unique challenges does it present? Is there meaningful signal in the benchmark, or is it just noise? The authors should discuss whether the benchmark contains realistic, non-trivial information and offer insights into its significance.
>
> **Response**: We are certain that there are meaningful signals in the dataset and have conducted analyses to demonstrate this. `Section 2.4` describes how we generate random embeddings to act as a sanity check baseline, expressly done to ensure the models are extracting real signal from the data. All of our results tables (`Tables 2`, `3`, and `4`) show performance metrics computed with the random embeddings to show the performance floor resulting from entirely uninformed embeddings. In these tables, one can see that all models do recover a real, useful signal from the data and exceed the performance floor.
>
> The benchmark is realistic as these are real astronomical time series captured by research-grade telescopes. It is also non-trivial to extract information from the light curves as our Figure 1 demonstrated, light curves have unique challenges that are yet to be addressed by existing TSFMs. Furthermore, the tasks that we evaluate these models on are specifically chosen to maximize the scientific value we can provide to the astrophysics community.

---

> ### Author Response · Authors · 2025-11-22
> **Part 4 of 4: Benchmarking Additional TSFMs**
>
> > Additionally, including more baseline models for comparison such as Toto, Moment, or PatchTST would strengthen the paper’s contribution and provide a more comprehensive evaluation.
>
> > It would be beneficial to test the benchmark on other foundational models from the literature, as well as models commonly used in time series forecasting, to provide a more comprehensive evaluation.
>
> **Response**: Thank you for this suggestion. Our goal in StarEmbed is not to exhaustively benchmark every existing TSFM, but to answer a focused question: can state-of-the-art general-purpose TSFMs, trained with the two dominant pretraining paradigms, transfer effectively to astronomy? To this end, we deliberately select Chronos and Moirai because (i) they are, to the best of our knowledge, among the strongest and most widely used open-source TSFMs, and (ii) they instantiate the two representative training tracks for foundation models: masked-token prediction (BERT-like) and autoregressive next-token prediction (GPT-like). Including additional TSFMs that are earlier generations or empirically weaker than these models would significantly increase computational and presentation overhead without providing new qualitative insight beyond what Chronos and Moirai already reveal.
>
> Regarding “models commonly used in time series forecasting,” our main focus is specifically on zero-shot pre-trained TSFMs whose embeddings can be directly applied to downstream tasks without task-specific training. Adding many additional forecasting models that are not pre-trained foundation models and instead require supervised training separately on each dataset would be conceptually opposite to this goal: such models do not test cross-domain transfer of foundation models, but rather conventional task-specific training. We already compare against a diverse set of strong non-foundation baselines and domain-specific methods, covering the main model families used in practice (e.g., classical architectures and recent deep models tailored to astronomical light curves), which serve as the appropriate reference point; expanding that set further would be largely orthogonal to our central question and would not alter the key conclusions drawn from our current, representative baselines.
>
>
>
> ---
>
> Thank you again for the valuable feedback. We hope our clarification addresses your concerns and welcome further discussion!

---

### Official Review · Reviewer_bckc · 2025-11-01

**Soundness:** 3
**Presentation:** 2
**Contribution:** 2
**Rating:** 6
**Confidence:** 4

**Summary:**

This paper introduces a benchmark for evaluating time series foundation models (TSFMs) on astronomical light curves from periodic variable stars. The authors curate ~40k labeled light curves from ZTF across seven astrophysical classes and evaluate several TSFMs, a domain-specific model (Astromer), and hand-crafted features on three tasks: unsupervised clustering, supervised classification, and out-of-distribution (OOD) detection.

**Strengths:**

1. Novel scientific benchmark -- scientific datasets have nonstandard and interesting properties (as highlighted in an intro figure -- heteroskedacticity, irregularity, etc), and are a good stress test of ML models. The paper has a nice introduction of astro time series data for non-experts.
2. Overall a careful, comprehensive benchmark -- The three-task evaluation framework (clustering, classification, OOD detection) provides a thorough assessment of model capabilities of different types. The use of expert labels, careful data cleaning and stratified splitting procedures are well-designed.
3. Good reproducibility / data availability (expected for datasets and benchmarks track paper)
4. Findings are interesting, in particular the generalizability of general-purpose TSFMs to astro data.

**Weaknesses:**

1. Literature review could be a bit more thorough, and the paper put in broader astro and time series FM context. E.g., https://arxiv.org/abs/2504.20290, https://arxiv.org/abs/2408.16829 and https://arxiv.org/abs/2405.17156 are some relevant papers, not necessarily directly on transients, but nevertheless applicable to supernova time series science. https://arxiv.org/abs/2405.13867 is relevant as well.
2. As far as I can tell there is only zero shot performance. One/few shot results would be extremely relevant given foundation model context.
3. Stats rigour -- only 3 seeds for non-deterministic methods seems insufficient, confidence intervals not universal
4. Comparison mostly at the performance level. For astro, speed is often paramount (in particular for transient science

**Questions:**

1. Can you provide any analysis of which hand-crafted features drive the strong baseline performance? Could help understand what TSFMs might be missing.
2. Would you expect results to change substantively in the 1/few shot setting?
3. Are general purposed TSFMs more costly/slower than astro specific ones?
4. Overall, it's hard to tell how comprehensive the comparison to astro methods is. There are lots of other methods (e.g. template fitting, other NN based methods that are referenced but not compared). I would be curious to what extent some of these are relevant or not.

---

> ### Author Response · Authors · 2025-11-22
> **Part 1 of 4: Comments about Literature Review**
>
> Thank you for taking the time to review our submission and for providing thoughtful feedback. We appreciate your efforts and the opportunity to clarify several points raised in your review. The following is our response, and the revision includes corresponding updates (marked in BLUE).
> > Literature review could be a bit more thorough, and the paper put in a broader astro and time series FM context.
> > E.g., https://arxiv.org/abs/2504.20290, https://arxiv.org/abs/2408.16829 and https://arxiv.org/abs/2405.17156 are some relevant papers, not necessarily directly on transients, but nevertheless applicable to supernova time series science. https://arxiv.org/abs/2405.13867 is relevant as well.
>
> **Response**:
>
> We'd firstly like to clarify one astronomy-specific point. Our study is focused on variable stars, e.g., pulsating stars which show periodic variations in their brightness. Supernovae and transients are another category of astrophysical phenomena that pertain to the terminal explosions of stars. Domain-specific models are not designed to work across these boundaries because these phenomena have very different time series. For instance, the time series of transients/supernovae look roughly bell-shaped and typically last for weeks to months, while the time series of variable stars, like those in this work, show periodic patterns continuing for millennia. Nevertheless, there is plenty to learn from work on time series of supernovae/transients for our work on variable stars.
>
> Maven (Zhang et al. 2024; arXiv:2408.16829) is certainly an important contribution in the astronomy foundation model and time series literature and is already cited in the text (`Section 2.2`, `line 154 - 171`). This model does not offer an appropriate comparison to the TSFMs we benchmark in this study because Maven is focused solely on supernovae, **a different category of astrophysical phenomena than the variable stars we study here.** FALCO (Zuo et al. 2025; arXiv:2504.20290) is similarly an important contribution to this area and is also already cited and discussed in the text (`Section 2.2`, `line 154 - 171`). The published version of FALCO is designed to only work on time series from the Kepler space telescope. As discussed in `Section 2.2`, this is a much more limited domain than the Astromer models (Donso-Oliva et al. 2023, 2025; arXiv:2205.01677, 2502.02717), which are designed to work on light curves from any observatory. For this reason, we cite but do not benchmark FALCO against the TSFMs.
>
> The two scaling law studies you share are valuable works guiding the development and growth of the TSFM field. We have added citations to these two works to the text. We do clarify here that the Scaling Law for Stellar Light Curves study, although pioneering it the area of scaling laws, does not consider the full breadth of challenges typical of variable star light curves nor does it explore different or pre-trained model architectures for downstream tasks as we do here. Their study (i) uses a small, special subset of light curves which are regularly sampled and (ii) only considers training a GPT-2 model from scratch. In our study, **we introduce more challenging and more common light curves as well as a much greater breadth of model architectures with very strong pre-training regimens.**

---

> ### Author Response · Authors · 2025-11-22
> **Part 2 of 4: Discussion on Zero/One/Few Shot Performance**
>
> > As far as I can tell there is only zero shot performance. One/few shot results would be extremely relevant given foundation model context.
>
> > Would you expect results to change substantively in the 1/few shot setting?
>
> **Response**: We thank the reviewer for this suggestion. Our goal in this work is to answer a very specific first-order question: *how useful are existing, large TSFMs without any model adaptation for real, irregular astrophysical light curves?* This is fundamentally a question of generalization capabilities, motivated by strong claims made following the introduction of the latest-generation TSFMs (Chronos, Chronos-bolt, Moirai). To this end, we explicitly treat these TSFMs as frozen embedding models and do not fine-tune them on our data. This “zero-shot” setting on new, out-of-domain data is precisely where generalization abilities can be tested.
>
> Methodologically, we **do go beyond pure zero-shot prediction** by training lightweight supervised heads (k-NN, linear probe, RF, MLP) on top of frozen embeddings (`section 4.2`, `line 349 - 366`). This isolates the question we care about: do the representations produced by TSFMs, without architecture- or domain-specific fine-tuning, perform competitively with conceptually very simple baselines? Our results show that they do: TSFMs approach or surpass these baselines for clustering and OOD detection and come close for supervised classification, despite never being trained on astronomical data. This alone is a strong and, we believe, surprising conclusion.
>
> To better understand the potential of adaptation, we also conducted a preliminary, deliberately simple fine-tuning experiment on Moirai-small, which we also describe in `Appendix G` of the appendix of the revision of the paper (`line 1329 - 1378`). We fine-tuned the model on ~1M ZTF astrophysical light curves for about 24 hours using the same masked token prediction task as in the pre-training. The training loss decreased from ~1.25 to ~1.15 within the first ~20k steps but then entered a regime of strong fluctuations around a saturated plateau over the next ~180k steps, without clear signs of further improvement. We view this as evidence that straightforward, off-the-shelf fine-tuning is not trivial and does not immediately yield clear gains over the zero-shot setting, likely due to the significant gap between the irregular, sparse astrophysical light curves and the regular time series on which these TSFMs were originally pre-trained. Our intention is not to claim that fine-tuning is impossible, but rather that it is a nontrivial, model- and domain-design problem in its own right.
>
> Exploring one/few-shot model adaptation (e.g., careful parameter-efficient fine-tuning of selected layers under small labeled budgets, with appropriate regularization and class balancing) would therefore require substantial additional design choices: which layers to adapt, which PEFT strategy, what label budgets, how to avoid overfitting on common classes, etc. This constitutes a large and interesting future research track that is beyond the scope of our current benchmark paper. **Our contribution here is to establish StarEmbed and show that zero-shot TSFMs already achieve surprisingly strong performance, thereby motivating and de-risking such follow-up work.**
>
> We do view few-shot and full fine-tuning as an important next step that our benchmark is explicitly designed to support: any future work can take our public StarEmbed dataset and evaluation pipeline and plug in one/few-shot–adapted TSFMs or other architectures. Furthermore, the LLM-style one/few-shot in-context learning regime, where no model parameters are updated and the few labeled examples are only provided as part of the input, is not currently feasible on existing TSFMs without substantial architectural changes. Existing TSFMs are designed as autoregressive forecasters, not as models that condition on support examples in a prompt-like context, so again, this would distract from our goal of evaluating the generalization capabilities of current TSFMs.

---

> ### Author Response · Authors · 2025-11-22
> **Part 3 of 4: Addressing Minor Questions**
>
> > Stats rigour -- only 3 seeds for non-deterministic methods seems insufficient, confidence intervals not universal
>
> **Response**: Thank you for the feedback. We agree that more runs would make the statistics more informative. We increased all experiments to use **10 random seeds**, and the resulting metrics are updated in our revised manuscript. The results do not change substantially, but are certainly more robust because of the large quantity of runs. The places in which we do not report confidence intervals are where methods are deterministic and thus will not vary across successive runs.
>
>
> > Comparison mostly at the performance level. For astro, speed is often paramount (in particular for transient science
>
> **Response**: Indeed, a model's computational cost is generally an important consideration. In these scenarios, real astrophysical analysis happens only after significant filtering has been done on the data, so the computational cost is not a major concern. We also note here that the data we are concerned with are variable stars and not transients/supernovae. In transient science, latency imposed by inference is very important as certain phenomena can become prohibitively difficult to measure after timescales of hours to days. The same is not true for variable star,s which persist for thousands of years.
>
> > Can you provide any analysis of which hand-crafted features drive the strong baseline performance? Could help understand what TSFMs might be missing.
>
> **Response**: You are absolutely correct that we should include a discussion of which hand-crafted features are most important for driving their impressive performance and that insights from such an analysis can help reveal why TSFMs are underperforming. We have conducted this analysis and described its results to reviewer bdhm in the paragraph with the heading "**Why hand‑crafted features are strong**". This analysis finds that the periods of variability computed from the "g" and "r" variates are the most important features. In the paragraph we reference, we describe how this result is aligned with previous studies and with astrophysical intuition. Also see the new `Appendix F` discussing this analysis and its results. The paragraph in response to a similar comment from `reviewer bdhm` with the heading "**Hypothesis on the shortcomings of current TSFMs**" ties this result to the underperformance of the TSFMs. In that paragraph, we describe how TSFMs lack the training data and the architectural capability to consider the uneven sampling of this astronomical time series data. Making this choice means TSFMs destroy the morphology of variations in the astronomical time series when they index measurements by their order rather than the actual time at which they were made. We assert that this shortcoming (in data and in architecture) is a driving reason for why TSFMs perform worse than hand-crafted features on the classification task.
>
>
> > Are general purposed TSFMs more costly/slower than astro specific ones?
>
> **Response**: Thanks for the question. The TSFMs in the benchmark is larger (10-15 M parameters) than the Astronomer model (up to 5 M parameters). All these models are transformer-based without mixture of experts mechanisms, hence the cost\inference time is proportional to the model parameters. Thus, the additional computational cost of the TSFMs is small, and since we are not concerned about latency, the inference time is not a concern either.

---

> ### Author Response · Authors · 2025-11-22
> **Part 4 of 4: Discussion on Comprehensiveness of Baselines**
>
> > Overall, it's hard to tell how comprehensive the comparison to astro methods is. There are lots of other methods (e.g. template fitting, other NN based methods that are referenced but not compared). I would be curious to what extent some of these are relevant or not.
>
> **Response**: Thank you for raising these important points. Template fitting is mostly used for the extraction of astrophysical stellar parameters from light curves. There is some limited work that uses goodness-of-fit statistics on matches to templates as input to a tree-based classifier (e.g., Sesar et al. 2017; arXiv:1611.08596), although this is still done alongside traditional hand-crafted feature extraction. Critically, estimating the variability period, the most important stage of hand-crafted feature extraction, is a prerequisite to template fitting. We have added a mention and citation of this technique to the text.
>
> Indeed other neural network-based methods have been explored for variable star analysis. Recurrent neural networks (RNNs), in particular, once received significant attention in the variable star literature. Despite extensive work tweaking the model design using long short-term memory (LSTM) architectures, gated recurrent units (GRUs), bi-directional RNNs and variants of each, RNNs have not yielded performance clearly surpassing that of hand-crafted features. GRUs have been the most popular and most performant; in `Section 2.1` we have cited Muthukrishna et al. 2019 (arXiv:1904.00014), Becker et al. 2020 (arXiv:2002.00994), and Shah et al. 2025 (arXiv:2501.01496). We also cite Naul et al. 2018 (arXiv:1711.10609) which compares the performance of hand-crafted features versus a sophisticated bi-directional autoencoding RNN and finds that for 2/3 tests they yield the same performance within 1-sigma uncertainties. As discussed in `Section 2.1`, for this reason we consider hand-crafted features our baseline and do not benchmark RNNs.

---

### Official Review · Reviewer_bdhm · 2025-11-01

**Soundness:** 2
**Presentation:** 3
**Contribution:** 1
**Rating:** 0
**Confidence:** 4

**Summary:**

The paper introduces an astronomy-focused benchmark for time series classification and regression. It does not propose new modeling methodology, but rather evaluates existing domain-specific deep time series models (e.g., Astromer, Chronos). The empirical finding that hand-crafted features outperform deep models across most tasks is interesting, and highlights a real and current limitation of existing “foundation” time series models in astronomy.

However, the paper stops at this observation. It does not provide a clear hypothesis, insight, or direction on why this is the case or how future time series models might be improved. As a result, the contribution feels limited to benchmarking — the results themselves are somewhat inconclusive, and the broader impact beyond the domain remains unclear.

**Strengths:**

* The paper is clearly compiled with in-depth knowledge of astronomy and the current state-of-the-art in deep learning models for astronomical time series data
* The experiments seem to span a variety of astronomic time series tasks
* The paper is very extensive, also including the appendix analyses. It clearly captures a large amount of work, which I think it would be better appreciated (and reviewed with more domain knowledge) in a domain-specific journal rather than ICLR.

**Weaknesses:**

* inconclusive results: hand-crafted features are best across most tasks, but no hypothesis is given how to improve the current models. Also no model or training algorithm is proposed
* the paper seems to miss a body of work around the time series classification and benchmarking community, mainly the UCR archive and the time series models benchmarked on these very diverse tasks. Including these non-deep learning models as comparisons may fill a gap between underperforming deep models and the very well performing hand-crafted features

**Questions:**

* Beyond deep time series classification, how applicable are the models tested in the time series classification community that are benchmarked e.g., on the UCR Time Series Classification Archive (https://www.cs.ucr.edu/%7Eeamonn/time_series_data_2018/) for these tasks? Given that hand-crafted features are performing well, these non-deep learning models from UCR may also be fairly competitive on the benchmarks
* Would state-space-models like Mamba also be applicable in this area of applications?

---

> ### Author Response · Authors · 2025-11-22
> **Part 1 of 6: Analysis of Hand-crafted Feature Performance**
>
> Thank you for taking the time to review our submission and for providing thoughtful feedback. We appreciate your efforts and the opportunity to clarify several points raised in your review. The following is our response, and the revision includes corresponding updates (marked in BLUE).
>
> > The empirical finding that hand-crafted features outperform deep models across most tasks is interesting, and highlights a real and current limitation of existing “foundation” time series models in astronomy. However, the paper stops at this observation. It does not provide a clear hypothesis, insight, or direction on why this is the case.
>
> **Response**: We perform an additional analysis investigating which of the hand-crafted features are most important for classification. This is done using the feature importance metric computed by assessing how performance degrades when that feature is excluded from random forest classifier. We find that the period is clearly the most important feature for classification. This finding is in-line with previous works on variable star classification with tree-based models (Richards et al. 2011; arXiv:1101.1959, Dubath et al. 2011; arXiv:1101.2406, Kim et al. 2015; arXiv:1512.01611). This also aligns with astrophysical expectations. We add a description and interpretation of this additional test to the text as a new appendix section (`Appendix F`) with a new figure.
>
> This finding is also strong support for our interpretation of the shortcomings of current TSFMs, described in Part 3.

---

> ### Author Response · Authors · 2025-11-22
> **Part 2 of 6: Insights into TSFMs and Directions for Improvement**
>
> >  It does not provide a clear hypothesis, insight, or direction on why this is the case or how future time series models might be improved.
>
> **Response**:
>
> **Hypothesis on the shortcomings of current TSFMs.** Current general-purpose TSFMs (Chronos, Chronos-bolt, Moirai) are trained on regularly sampled time series which are usually uni-variate and without any measurement uncertainties. The existing architectures do not have specific components to tackle all of these challenges characteristic of astronomical and many other scientific time series. We isolate irregular sampling and the gappy nature of astronomical time series to visualize this weakness, **see Figure 4 in the revision.** The figure shows a real light curve in our data set where measurements and indexed by the time of the observation (the real, physical index) and indexed by the order in which they were taken (the non-physical index which **current TSFMs "see" as a result of not properly treating the irregular sampling**).
>
> Without examples of time series with at least some of these challenges in the training data and the proper architectural mechanisms for handling them, TSFMs will fail to perfectly represent the temporal evolution of the time series, placing a ceiling on their ability to encode important features, such as the period. We posit that this shortcoming is precisely the reason why TSFMs underperform the basic, classical method of hand-crafted feature extraction on most tasks in our benchmark. **The additional hand-crafted feature importance analysis we describe above reveals that the period of the variability is the single most important feature in conducting classification.** This new figure very clearly illustrates the destruction of this critical feature in the TSFMs' "view" of the time series. We have added discussion of these TSFM shortcomings and this new figure to the text.
>
> To further test this hypothesis, **we have added period regression as an additional test**:
>
>
> | Model            | RMSE | R^2 |
> | ---------------- | :------: | :-------------: |
> | MOIRAI       | 26.355 $\pm$ 0.022|0.295 $\pm$ 0.001|
> | Chronos‑bolt | 23.414 $\pm$ 0.062|0.443 $\pm$ 0.002|
> | Chronos      | 23.178 $\pm$ 0.076|0.455 $\pm$ 0.003|
> | Astromer‑1   | 33.272 $\pm$ 0.015|0.005 $\pm$ 0.0009|
> | Random‑emb   | 33.452 $\pm$ 0.016|-0.005 $\pm$ 0.0009|
>
> While differences do exist in the regression performance of these embeddings (see Chronos vs Astromer-1), it is important to note that they all perform significantly worse than the Lomb Scargle period finding algorithm which is a reliable, albeit computationally expensive, method used for period finding in astronomy--and is used in the hand-crafted features baseline. **This further supports our above hypothesis.**
>
> *Note:* We exclude handcrafted features from the period regression test since they already contain the period.
>
>
> A promising direction for the improvement of current TSFMs is to (i) include time series data with challenges characteristic of astronomical time series (irregular sample, multiple variates, heteroskedastic uncertainties, regular and irregular gaps) and (ii) improve upon the architectural ability of the models to properly treat these challenges, for example, by designing positional encodings that correctly handle irregular sampling. Such work is different from our aims here, so we consider it beyond the scope of this study.
>
> **Astromer zero‑shot failure mode.**
> Unlike the general-purpose TSFMs, the astronomy-specific transformer-based models, the Astromer family, do have architectural components designed to handle irregular sampling and other challenges in this data. We find that the Astromer models fall short due to their poor generalization capabilities. Our analysis in Appendix E shows that Astromer-1's embeddings of our data collapse to a single direction. We interpret this and the good performance of the TSFMs in our benchmarks as a suggestion that the vast pre-training done in TSFM development, which is absent in the development of Astromer, has significant benefits in terms of generalization.

---

> ### Author Response · Authors · 2025-11-22
> **Part 3 of 6: Contribution of Our Benchmark**
>
> > The contribution feels limited to benchmarking ... and the broader impact beyond the domain remains unclear
>
> **Response**:
>
> **New type of time series never before seen by the ML community.** Our work provides the first data set of astronomical time series with high-quality expert labels that is prepared for use by AI practitioners. These do not appear in the large pretraining corpora used by TSFMs, thus they provide a unique test of TSFM generalization without any risk of data leakage (See `Section 4` and `Appendix F.2` of [1]). These data also expose characteristics that standard benchmarks do not capture: the data are highly irregularly sampled, have long and heterogeneous gaps, and the target variate has heteroskedastic measurement uncertainties. These properties are common in scientific time series, not just astronomy, but are rarely represented in current time series benchmark data sets. Our benchmark is designed to bridge this gap.
>
> **A diagnostic testbed for missing capabilities in TSFMs.** Hand-crafted features have been the literature-standard choice for variable star classification for decades. Moreover, the method has not evolved a great deal over the past decade--the same decade where deep learning and foundational AI have exploded in capabilities. In this work, **we are the first to pose the question of whether TSFMs can replace these pipelines.** This study, and the benchmark we introduce, provides **the first signals on where current TSFMs work well and where they fail on astronomical time series.** In brief, we find that TSFMs are SOTA or near-SOTA in all tasks but still fall short of hand-crafted features in the critical classification task. These results represent one of the best demonstrations of the **generalization abilities of TSFMs on the scientific time series data to date.** In the revised text, we have taken our analysis further to investigate why TSFMs fall short and show that, by not handling the irregular sampling, TSFMs destroy much of the information-rich features of the time series. This finding is a demonstration of how our benchmark will investigate the capabilities of current and future TSFMs on confidently out-of-domain data.
>
>
> **Role within the broader datasets & benchmarks effort.** There is growing recognition in the ML community that advancements require not only algorithmic innovations but also high-quality, well-documented and challenging benchmarks on standardized, open data sets. These benchmarks should also reflect real applications rather than convenient toy problems [2-6]. The importance of benchmarks with these properties to continued progress in the AI for time series field cannot be overstated. Thus, although our work does not introduce algorithmic innovations, its importance lies in the novelty of the benchmark we introduce. Our benchmark provides a scientifically grounded, hard-to-simulate time series testbed that has already exposed successes and failures of current TSFMs.
> **Furthermore, our organized, published dataset will dramatically lower the barrier to entry for ML researchers and practitioners to contribute
> to variable star science**.
>
> ---
>
> [1] Aksu, T., Woo, G., Liu, J., Liu, X., Liu, C., Savarese, S., ... & Sahoo, D. (2024). Gift-eval: A benchmark for general time series forecasting model evaluation.
>
> [2] Wang, Alex, et al. "GLUE: A multi-task benchmark and analysis platform for natural language understanding." ICLR (2019)
>
> [3] Fan, Jingxuan, et al. "Hardmath: A benchmark dataset for challenging problems in applied mathematics." ICLR (2025)
>
> [4] Zhang, Hanlei, et al. "Mintrec2. 0: A large-scale benchmark dataset for multimodal intent recognition and out-of-scope detection in conversations." ICLR (2024)
>
> [5] Nascetti, Andrea, et al. "Biomassters: A benchmark dataset for forest biomass estimation using multi-modal satellite time-series." NeurIPS (2023)
>
> [6] Cachay, Salva Rühling, et al. "ClimART: A benchmark dataset for emulating atmospheric radiative transfer in weather and climate models." NeurIPS (2021)

---

> ### Author Response · Authors · 2025-11-22
> **Part 4 of 6: Relation to UCR Archive and Non-Deep Learning Models**
>
> > The paper seems to miss a body of work around the time series classification and benchmarking community, mainly the UCR archive and the time series models benchmarked on these very diverse tasks. Including these non-deep learning models as comparisons may fill a gap between underperforming deep models and the very well-performing hand-crafted features.
>
> **Response**: We appreciate the reviewer’s suggestion to include models benchmarked on the UCR archive. The UCR Archive has been instrumental to the machine learning and time-series field, providing a unified, rigorously curated benchmark that researchers worldwide rely on to test, compare, and validate new time series prediction models. However, **our work addresses a fundamentally different question**: we evaluate whether large pre-trained time-series foundation models can generalize to astronomical, highly irregular, heteroskedastic light-curve data, a regime that is **not represented in the UCR archive.** As shown in our paper, astronomical observations contain multi-band measurements, irregular sampling, missingness, and heteroskedastic uncertainties, all of which differ substantially from the regular, fixed-length time series in UCR benchmarks.
>
> Benchmarking a broad suite of classical and deep learning methods on general time-series data has indeed been performed extensively in prior work. Our goal is instead to assess whether off-the-shelf foundation embeddings, without any fine-tuning, can match or exceed long-standing state-of-the-art astrophysical baselines based on hand-crafted features. As our results show, this comparison is already nontrivial and scientifically illuminating, and additional experiments, including those non-deep learning models as comparisons, would not change the conclusions nor address the core motivation of this study.

---

> ### Author Response · Authors · 2025-11-22
> **Part 5 of 6: Relevance of State-space Models to StarEmbed**
>
> > Would state-space-models like Mamba also be applicable in this area of applications?
>
> **Response**: Previous works on State-Space Models typically assume evenly spaced observations. In recent years, SSMs such as Mamba are a promising class of architectures for continuous/irregular time series modeling. By parameterizing a latent continuous dynamical system and using fast convolution-style implementations, they can model long-range temporal dependencies with good computational efficiency. In principle, they could be applied to the analysis of astrophysical data. Although still in its early age, prior work has shown results on challenging sequence tasks and several time-series tasks [1,2]. **However, these works still develop in a one-model-per-task trend.** Our focus in this work is not on proposing or comparing new sequence architectures trained from scratch, but on evaluating existing, large pre-trained time-series foundation models in a strict zero-shot setting on real astrophysical data. **At present, there does not exist a broadly pre-trained, publicly available Mamba-based TSFM that could be used in exactly the same “drop-in zero-shot embedding” fashion as Chronos or Moirai.** Applying Mamba would therefore require substantial additional model development and large-scale training, which is orthogonal to our main goal of assessing cross-domain transfer of current TSFMs.
>
> Conceptually, our benchmark is architecture-agnostic: any future foundation model, whether Transformer-, SSM-, or hybrid-based, could be evaluated on StarEmbed by producing embeddings for our light curves and plugging them into the same downstream heads and metrics. We thus view state-space models like Mamba as an interesting direction for future work, and our benchmark is designed precisely so that such models can be fairly compared once suitable pre-trained versions become available.
>
> ---
>
> [1] Gu, A., Goel, K., & Ré, C. (2021). Efficiently modeling long sequences with structured state spaces. arXiv preprint arXiv:2111.00396.
>
> [2] Zhang, M., Saab, K. K., Poli, M., Dao, T., Goel, K., & Ré, C. (2023). Effectively modeling time series with simple discrete state spaces.

---

> ### Author Response · Authors · 2025-11-22
> **Part 6 of 6: Clarifications Regarding the Summary**
>
> We appreciate your thoughtful summary and would like to offer a few clarifications to prevent possible misunderstandings about the scope and significance of our work. First, StarEmbed is not primarily a benchmark of “existing domain-specific deep time series models (e.g., Astromer, Chronos).” Astromer is indeed an astronomy-specific model and serves as one of our domain specific baselines, which our TSFMs are able to match or surpass. In contrast, Chronos and Moirai are general-purpose time series foundation models pre-trained on large, multi-domain corpora of time series spanning hundreds of datasets and application areas (e.g., retail, traffic, energy, IoT), not on astronomical data. Our central question is whether such large, cross-domain TSFMs can transfer to irregular astrophysical light curves without any further training of their parameters. Second, our results do not highlight a “limitation of existing foundation time series models in astronomy,” but almost the opposite: we find that zero-shot TSFM embeddings, combined with lightweight heads, are competitive with, and in some tasks surpass, strong domain-specific deep models such as Astromer and hand-crafted feature pipelines (see figure 3 for per-class performance), despite never seeing astrophysical data during pre-training. This is particularly important from a scalability perspective: using these models in a zero-shot way (i.e., without fine-tuning or repeatedly retraining a huge model on each new observatory) is what makes them attractive for the imminent petabyte-scale data sets from next-generation observatories. While our results also show that hand-crafted features remain very strong baselines, the main message is the surprising domain-transferability and practical promise of general-purpose TSFMs in astronomy, not their failure.

---

### Author Response · Authors · 2025-12-03
**Global Rebuttal Summary**

Dear Reviewers and Area Chairs:


We thank the reviewers for the hepful comments and detailed reviews. Broadly, three main concerns are raised. Below we summarize how the revision addresses each, giving the Area Chairs a quick overview of the reviews and responses. Details appear in the individual rebuttal notes linked below and in the blue-highlighted changes in the revised manuscript.

---


**Concern I: “Only a benchmark” / lack of insight into why TSFMs underperform classical SOTA baseline.** [`bdhm`, `bckc`, `qUQx`]

**Revisions:**
* We add a **new `Figure 4`, supporting period-regression experiment (`Appendix H`), and accompanying discussion (`Section 6`)** that illustrates how applying current TSFMs to irregular, gappy sampling destroys periodic structure. Detailed discussion in [bdhm-Part2](https://openreview.net/forum?id=ZXRlSzHoDE&noteId=fe6sfB3ocB) and [bckc-Part3](https://openreview.net/forum?id=ZXRlSzHoDE&noteId=nY1DVxBEce).
* We conducted new **feature-importance analysis** of hand-crafted features (`Appendix F` and `Figure 7`). Detailed discussion in [bdhm-Part1](https://openreview.net/forum?id=ZXRlSzHoDE&noteId=gQ8fhbRWBg).

---

**Concern II: Lack of novel finetuning algorithms.** [`bdhm`, `qUQx`]

**Revisions:**
* We conducted a **new finetuning experiment** using Moirai-small on ~1M variavble star light curves in `Appendix G`. The loss improvement quickly saturates hence suggests naïve fine-tuning on irregular astromical time series is non-trivial. Detailed discussion in [bckc-Part2](https://openreview.net/forum?id=ZXRlSzHoDE&noteId=mdEMXy6aAU) and [qUQx-Part1](https://openreview.net/forum?id=ZXRlSzHoDE&noteId=kcQVp0csg8).

---

**Concern III: Limited statistical characterization of the data set / statistical rigor for classification results.** [`bckc`, `qUQx`]

**Revisions:**
* We added a **statistical analysis of the data set** at the beginning of `Section 3, line 235-252`, including a new `Figure 2` showing the broad distribution of sampling frequencies and measurement uncertainties, dramatically contrasted with data that is typically used with TSFMs. Detailed discussion in [qUQx-Part2](https://openreview.net/forum?id=ZXRlSzHoDE&noteId=3ssiaeDYkI).
*  We expanded **all stochastic experiments from 3 to 10 random seeds** and update the results accordingly (`Table 2-4`, `figure 3`, `figure 9-14`). Detailed discussion in [bckc-Part3](https://openreview.net/forum?id=ZXRlSzHoDE&noteId=nY1DVxBEce).

----

We hope these revisions address the reviewers' concerns and improve the overall quality of our paper.

---

### Note · Authors · 2026-02-03

**Comment:**

We note that this year's ICLR review process included procedural circumstances that differed from a typical submission. To make it easier for future readers to understand our work after the revision in the rebuttal, we summarize the additions incorporated during the rebuttal and the paper's scope.

* In response to reviewer questions, we provide **4 new experiments**: (1) feature-importance analysis of hand-crafted features, (2) new finetuning experiment on ~1M star light curves (3) statistical analysis of the data set, and (4) period-regression experiment. Additionally, all stochastic experiments were increased from 3 runs to 10 runs.

    These new experiments are presented in the rebuttal and revision but are not discussed in the meta-review.

*  We provide the first open-source and task-diverse benchmark of TSFMs on astrophysical data. The results show both their capabilities (e.g., top performance on Clustering/OOD detection compared to SOTA handcrafted features) and limitations (e.g., classification performance second to handcrafted features + period regression). The diversity of tasks provides a comprehensive evaluation for TSFMs on light curve analysis.
Furthermore, the insight regarding zero-shot performance of TSFMs will become increasingly important with upcoming surveys (e.g., LSST), as the community needs efficient ways to extract knowledge from massive peta-scale datasets.

We thank the reviewers and the area chair for their time, feedback, and engagement with the work. We withdraw the submission with this note so that future readers have a clear view of the expanded experimental evidence and the full scope of the benchmark developed in this work.

**Withdrawal Confirmation:**

I have read and agree with the venue's withdrawal policy on behalf of myself and my co-authors.

---

### Meta-Review · Area_Chair_DWRP · 2025-12-22

**Summary:**

This work presents the first unified evaluation benchmark for applying time-series foundation models to astrophysical light-curve classification. By systematically examining their generalization and transfer behavior on irregularly sampled and heteroscedastic data, the study offers valuable insights into both the strengths and limitations of current TSFMs in realistic astronomical settings. The manuscript is clearly structured and accessible, reflecting the authors’ strong expertise in astronomy as well as modern deep learning techniques for time-series analysis.

**Reviewer Concerns:**

All reviewers expressed substantial concerns regarding the technical contributions of the paper, noting that it does not introduce significant innovations in model architecture or algorithmic design. Reviewer bdhm pointed out that the baseline comparisons should include non–deep learning methods commonly used in the UCR archive. Reviewer bckc suggested that the experimental protocol should incorporate one-shot or few-shot evaluation settings, while reviewer qUQx emphasized the need for comparisons with a broader range of time-series foundation models, particularly methods such as MOMENT.

In response to these requests for additional experiments, the authors mainly argued that the data types are incompatible or that the proposed additions would not alter the paper’s conclusions. However, such responses are not sufficiently convincing.

Since the primary contribution of the paper is the introduction of a benchmark, the experimental design is expected to be as comprehensive and informative as possible. For example, non–deep learning baselines for UCR datasets could be included by applying preprocessing strategies such as interpolation or imputation to enable fair comparison with classical time-series classification methods. Moreover, MOMENT is also pre-trained on heterogeneous benchmark datasets, and several recent time-series foundation models, including TimeMoE (ICLR 2025), Sundial (ICML 2025), and Moirai-MoE (ICML 2025) for forecasting, as well as Nutime (TMLR 2024) and Mantis (arXiv 2025) for classification, would constitute relevant and informative additional baselines.

**Reviewer Scores:**

The four reviewers did not participate in the discussion. Reviewer bdhm assigned an overall score of 0, indicating that this score could potentially be raised to 2. Reviewer bckc gave an overall score of 6, which is unlikely to change. Reviewers qUQx and xzSZ both assigned an overall score of 4, and neither indicated a likelihood of score revision.

---

### Decision · Program_Chairs · 2026-01-26

Reject